# Image Background Serves as Good Proxy for Out-of-distribution Data

**Sen Pei**
ByteDance Inc.
`peisen@bytedance.com`

## Abstract

Out-of-distribution (OOD) detection empowers the model trained on the closed image set to identify unknown data in the open world. Though many prior techniques have yielded considerable improvements in this research direction, two crucial obstacles still remain. Firstly, a unified perspective has yet to be presented to view the developed arts with individual designs, which is vital for providing insights into future work. Secondly, we expect sufficient natural OOD supervision to promote the generation of compact boundaries between the in-distribution (ID) and OOD data without collecting explicit OOD samples. To tackle these issues, we propose a general probabilistic framework to interpret many existing methods and an OOD-data-free model, namely **S**elf-supervised **S**ampling for **O**OD **D**etection (SSOD). SSOD efficiently exploits natural OOD signals from the ID data based on the local property of convolution. With these supervisions, it jointly optimizes the OOD detection and conventional ID classification in an end-to-end manner. Extensive experiments reveal that SSOD establishes competitive state-of-the-art performance on many large-scale benchmarks, outperforming the best previous method by a large margin, *e.g.*, reporting **-6.28%** FPR95 and **+0.77%** AUROC on ImageNet, **-19.01%** FPR95 and **+3.04%** AUROC on CIFAR-10, and top-ranked performance on hard OOD datasets, *i.e.*, ImageNet-O and OpenImage-O.

## 1 Introduction

Identifying the out-of-distribution (OOD) samples is vital for practical applications since the deployed systems can not handle these unknown data in a human-like fashion, *e.g.*, performing rejection. Based on research in animal behavior, such as Tierney & Jane (1986) and Shettleworth (2001), these abilities demonstrated by humans are rooted in experience, which means we earn them from interactions with environments. For instance, the adults will not reach their fingers towards the flames in case of burning, while the children may take this risky action due to their limited cognitive ability. This principle reveals that the failure encountered by the well-trained classification models in detecting OOD examples results from the lack of interactions with OOD data. Consequently, acquiring sufficient and diversified OOD signals from the in-distribution (ID) data has become a promising direction, which helps mitigate the gap between classification and OOD detec-

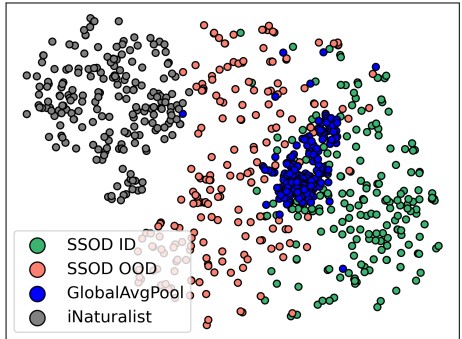

Figure 1: Feature visualization of the ID and OOD images. The green/orange dots are surrogate ID/OOD features generated by SSOD. The blue/gray dots are natural ID/OOD features of ImageNet/iNaturalist.

tion. Fig. 1 provides an intuitive illustration of natural supervision's importance in OOD detection. In particular, we employ the pre-trained ResNet-50 (He et al., 2016) as the backbone and ImageNet (Russakovsky et al., 2015) as the ID data. The OOD samples are from iNaturalist (Horn et al., 2018). The blue points are pooled features from ImageNet, *i.e.*, the natural ID features. The green/orange points indicate the ID/OOD signals generated by our SSOD, *i.e.*, the ID/OOD proxy. We can see that

the OOD signals sampled by SSOD build a defensive wall between the real ID and OOD data, which can serve as the proxy of authentic OOD images to supervise the training procedure.

Recall other existing OOD detection methods that primarily rely on the perspective of statistical difference, *i.e.*, observing distinctions of the pre-trained features between ID and OOD samples. These methods use heuristic rules to filter out OOD data in a two-stage manner, *i.e.*, pre-training and post-processing, which suffer from the following drawbacks. Firstly, the frozen model weights are obtained on the ID classification task with limited OOD supervision. Therefore, the extracted features inherently carry bias, which is not distinguishable enough for identifying OOD data (cf. Fig. 4). Secondly, the two-stage design yields poor scalability and efficiency since it is unsuitable for scenarios without pre-trained models, *e.g.*, given insufficient training data, a two-stage model may fail to obtain high-quality weights, thus encountering the performance drop in detecting OOD data.

To tackle the issues above, this paper interprets the OOD detection task with a unified probabilistic framework, which can widely include many previous individual designs. Concretely, our framework starts from Bayes' rule and divides the robust classification problem into two tasks: conventional ID classification and OOD detection. According to our theoretical analysis, the deficiency encountered by traditional neural networks in identifying OOD data arises from the absence of a critical component, *i.e.*, an OOD factor that estimates the likelihood of images belonging to the in-distribution. Furthermore, sailing from this general foundation, we present **S**elf-supervised **S**ampling for **O**OD **D**etection (**SSOD**), an end-to-end trainable framework **w/o** resorting to explicit OOD annotations. In contrast to the observed paths that hug synthetic OOD features, SSOD directly samples natural OOD supervision from the background of ID images, *i.e.*, self-supervised, getting rid of the constraints resulting from the lack of labeled OOD data and the deviation introduced within the OOD feature syntheses stage. Extensive experiments demonstrate that the joint end-to-end training significantly improves the OOD detection performance and guides the model to focus more on the object-discriminative characters instead of the meaningless background information (cf. Fig. 2). The major contributions of this paper are summarized as follows:

- We establish a general probabilistic framework to interpret the OOD detection, where various OOD methods can be analyzed, with main differences and key limitations clearly identified.

- To promote ID/OOD features being more distinguishable, we design an end-to-end trainable model, namely **S**elf-supervised **S**ampling for **O**OD **D**etection (**SSOD**), to sample natural OOD signals from the ID images. SSOD avoids the labor-intensive work of labeling sufficient OOD images.

- SSOD is evaluated across various benchmarks and model architectures for OOD detection, where it outperforms current state-of-the-art approaches by a large margin, *e.g.*, improving KNN (Sun et al., 2022) w/ and w/o contrastive learning with **-20.23%** and **-38.17%** FPR95 on Places (Zhou et al., 2018), and Energy (Liu et al., 2020) with **-30.74%** FPR95 and **+8.44%** AUROC on SUN (Xiao et al., 2010), to name a few. The scalability and superiority of SSOD promise its potential to be a starting point for solving the OOD detection problem.

## 2 METHODS

We introduce the unified probabilistic OOD framework, present the detailed interpretation of existing OOD detection methods from our unified view, and formulate the SSOD finally.

### 2.1 PROBABILISTIC OOD DETECTION

We formalize the OOD detection task as a binary classification problem. Concretely, we consider two disjoint distributions on the data and label space, denoted as $\mathcal{S}_{\mathbb{ID}} \times \mathcal{Y}_{\mathbb{ID}}$ and $\mathcal{S}_{\mathbb{OOD}} \times \mathcal{Y}_{\mathbb{OOD}}$, representing the ID/OOD distribution. We note that $\mathcal{Y}_{\mathbb{ID}}$ and $\mathcal{Y}_{\mathbb{OOD}}$ have no overlap, *i.e.*, $\mathcal{Y}_{\mathbb{ID}} \cap \mathcal{Y}_{\mathbb{OOD}} = \varnothing$. OOD detection aims to train a model that can effectively distinguish the source distribution of a given image $x$. Moreover, for $x \times y \in \mathcal{S}_{\mathbb{ID}} \times \mathcal{Y}_{\mathbb{ID}}$, the classifier $f(\cdot)$ should correctly predict its category.

Supposing that we have $M$ known classes within the ID data, depicting as $\{w_1, w_2, ..., w_M\}$. We don't distinguish the categories of OOD data. Thus, the images sampled from unknown classes are depicted as the $w_{M+1}$ no matter their categories or domains. Given an image $x$, we aim to learn a classifier $f(\cdot)$ that reports the posterior probability of $x$ belonging to each category, which is $P(w_i|x)$

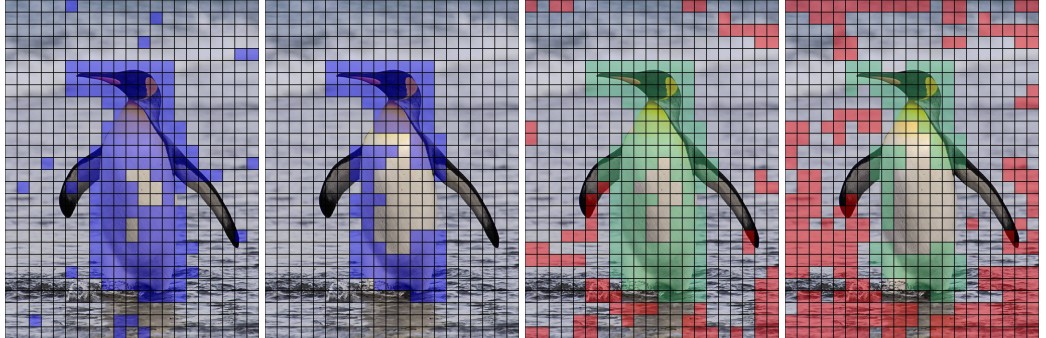

(a) Locality of ResNet-50.    (b) Locality of SSOD.    (c) Samplers of ResNet-50.    (d) Samplers of SSOD.

Figure 2: **(a)**: The image patches in blue are recognized as *penguins* with over 40% confidence by the vanilla ResNet-50. **(b)**: The SSOD counterpart of (a). We can see that SSOD focuses on more discriminative characters, reducing the false negatives. **(c)**: The image patches in green/red are sampled as ID/OOD supervision with over 95% confidence by ResNet-50. **(d)**: The SSOD counterpart of (c), we notice that SSOD identifies more background as OOD data. SSOD is motivated by the local property of conventional networks as illustrated in (a), and the results depicted in (b) reveal that the joint training enhances the local property. The confidence is manually selected.

for $i \in \{1, 2, ..., M\}$. Noting that $P(w_{M+1}|x) = 1 - \sum_{i=1}^{M} P(w_i|x)$. We first consider $M$ binary classification problems, and correspondingly, we have $M$ discriminative functions to verify whether the input image $x$ belongs to category $w_i$. Taking $g_i(x) = -s_i + T$ as the score function, where $s_i$ is learnable with respect to the classifier and $T$ is the bias term. Greater $g_i(x)$ yields higher confidence of $x$ belonging to category $w_i$. To make the confidence meet intuition, we employ $\sigma(\cdot)$ function to map the logits $g_i(\cdot)$ in the form of probability, saying $P^b(w_i|x) = \sigma(g_i(x))$, where the superscript $b$ indicates the **b**inary classifier. With these auxiliary notations, we have the following formula:

$$P^b(w_i|x) = \sigma(g_i(x)) = \frac{1}{1 + e^{s_i - T}} \tag{1}$$

Based on the DS evidence theory (Dempster, 1967), *i.e.*, DST, we calculate the posterior probability of $x$ belonging to any ID category $w_i$ as follows:

$$P(w_i|x) = \frac{1}{Z} \cdot P^b(w_i|x) \cdot \prod_{j=1, j \neq i}^{M} (1 - P^b(w_j|x)) \tag{2}$$

where $Z$ is a normalization factor. Based on the equation that $P(w_{M+1}|x) = 1 - \sum_{i=1}^{M} P(w_i|x)$, we can obtain the expression of $Z$ as follows:

$$Z = \underbrace{\sum_{i=1}^{M} [P^b(w_i|x) \cdot \prod_{j=1, j \neq i}^{M} (1 - P^b(w_j|x))]}_{\text{ID Confidence}} + \underbrace{\prod_{j=1}^{M} (1 - P^b(w_j|x))}_{\text{OOD Confidence}} \tag{3}$$

In the equation above, the first term of $Z$ indicates the sum of probability that $x$ belongs to any known classes, while the last term represents that of $x$ is OOD data. We substitute the expression of $Z$ into Eq 2 for simplification. The obtained results are depicted as follows:

$$P(w_i|x) = \frac{e^{-s_i + T}}{1 + \sum_{j=1}^{M} e^{-s_j + T}} \tag{4}$$

We assume that $T$ is $s_{M+1}$ since they are both trainable. Concerning that $e^{-s_{M+1} + T} = e^0 = 1$, then, Eq 4 can be transformed as:

$$P(w_i|x) = \frac{e^{-s_i + T}}{\sum_{j=1}^{M+1} e^{-s_j + T}} = \frac{e^{-s_i}}{\sum_{j=1}^{M+1} e^{-s_j}} \tag{5}$$

The above expression of $P(w_i|x)$ is in the form of softmax classification, except the denominator contains an extra term, *i.e.*, $e^{-s_{M+1}}$. With simple post-processing transformation, Eq 5 can be depicted as follows:

$$P(w_i|x) = \frac{e^{-s_i}}{\sum_{j=1}^{M+1} e^{-s_j}} = \underbrace{\frac{e^{-s_i}}{\sum_{j=1}^{M} e^{-s_j}}}_{\text{ID Classification}} \cdot \underbrace{\frac{\sum_{j=1}^{M} e^{-s_j}}{\sum_{j=1}^{M+1} e^{-s_j}}}_{\text{OOD Detection}} \tag{6}$$

Clearly, the first factor of Eq 6 is the conditional probability that $x$ belongs to category $w_i$ assuming $x$ is sampled from ID data simultaneously. The second factor of the above equation indicates the probability that $x$ is ID images. Recall that $\mathcal{S}_{\mathbb{ID}}$ and $\mathcal{S}_{\mathbb{OOD}}$ are the sets of in-distribution and out-of-distribution data, thus, Eq 6 tells us a conclusion that:

$$P(w_i|x) = \frac{e^{-s_i}}{\sum_{j=1}^{M} e^{-s_j}} \cdot \frac{\sum_{j=1}^{M} e^{-s_j}}{\sum_{j=1}^{M+1} e^{-s_j}} \triangleq \underbrace{P(w_i|x \in \mathcal{S}_{\mathbb{ID}}, x)}_{\text{ID factor}} \cdot \underbrace{P(x \in \mathcal{S}_{\mathbb{ID}}|x)}_{\text{OOD factor}} \tag{7}$$

Since the output of conventional neural networks activated by $\mathrm{Softmax}$ is $P(w_i|x \in \mathcal{S}_{\mathbb{ID}}, x)$, we focus on formulating the second term (**OOD factor**), *i.e.*, $P(x \in \mathcal{S}_{\mathbb{ID}}|x)$.

## 2.2 Revisit OOD detection methods from the probabilistic view

We interpret several classic OOD detection techniques from the perspective of our proposed probabilistic framework and find that most OOD detection methods hold $P(w_i|x \in \mathcal{S}_{\mathbb{ID}}, x) = f_i(x)$, *i.e.*, the $i$-th dimension of the classifier's output activated by $\mathrm{Softmax}$ function. Thus, the crucial point is how to compute the OOD factor $P(x \in \mathcal{S}_{\mathbb{ID}}|x)$.

**Methods based on logits**, *e.g.*, Max-Softmax Probability (MSP) (Hendrycks, 2017), which directly employs the $\mathrm{Softmax}$ output of classifiers as the ID/OOD score, aiming to distinguish them with classification confidence. Concretely, given image $x$, MSP uses the following expressions to depict the procedure of OOD detection:

$$x \to f(\cdot) \to \begin{cases} x \in \mathcal{S}_{\mathbb{OOD}}, & \max f(x) < \gamma \\ x \in \mathcal{S}_{\mathbb{ID}}, & \max f(x) \geq \gamma \end{cases}. \tag{8}$$

MSP expects the classifier $f(\cdot)$ to assign higher confidence, *i.e.*, $\max f(x)$, to ID samples while lower of that to the OOD. Obviously, for MSP, the OOD factor is built as:

$$P(x \in \mathcal{S}_{\mathbb{ID}}|x) = P(\max f(x) \geq \gamma). \tag{9}$$

**Methods based on features** try to distinguish ID/OOD data based on their deep features extracted by the backbone $h(\cdot)$, such as ReAct (Sun et al., 2021), BAL (Pei et al., 2022), VOS (Du et al., 2022), and KNN (Sun et al., 2022), *etc*. Taking ReAct (Sun et al., 2021) as an example, it builds the OOD factor $P(x \in \mathcal{S}_{\mathbb{ID}}|x)$ in a hard threshold manner with linear projection, depicted as:

$$P(x \in \mathcal{S}_{\mathbb{ID}}|x) = P(\mathbf{W}^\top \mathrm{ReAct}(h(x), c) + \mathbf{b} \geq \gamma), \tag{10}$$

where $\mathbf{W}^\top$ and $\mathbf{b}$ are the weight matrix and bias vector, $\mathrm{ReAct}(h(x), c) = \min\{h(x), c\}$ is an element-wise truncation function dominated by the threshold $c$, and $\gamma$ is a hard threshold. Instead of the OOD-syntheses-free schemes like ReAct (Sun et al., 2021) and KNN (Sun et al., 2022), BAL (Pei et al., 2022) and VOS (Du et al., 2022) generate ID/OOD supervision in the feature space to optimize the ID/OOD classifier with $P(x \in \mathcal{S}_{\mathbb{ID}}|x) = \sigma(d(h(x)))$, where $d(\cdot)$ is a discriminator that expect to assign higher confidence for ID features.

In summary, most OOD methods approximate the OOD factor $P(x \in \mathcal{S}_{\mathbb{ID}}|x)$ by $P(f(x) \in f(\mathcal{S}_{\mathbb{ID}})|x)$ as MSP (Hendrycks, 2017), or $P(h(x) \in h(\mathcal{S}_{\mathbb{ID}})|x)$ like ReAct (Sun et al., 2021), BAL (Pei et al., 2022), VOS (Du et al., 2022), and KNN (Sun et al., 2022). We note here that $f(x)$ is the $\mathrm{Softmax}$ output, and $h(x)$ is the feature extracted by the backbone. Nevertheless, there is a significant bias introduced by $f(\cdot)$ and $h(\cdot)$ since they are trained for the ID classification. It adversely affects the discrimination of ID and OOD data. Furthermore, the generated OOD signals in feature space, *e.g.*, BAL (Pei et al., 2022) and VOS (Du et al., 2022), do not necessarily lead to the existence of a corresponding natural OOD image. Consequently, its effectiveness in open world scenarios may be limited. To remove these obstacles, we propose to sample OOD supervision from the ID images and optimize the OOD factor directly.

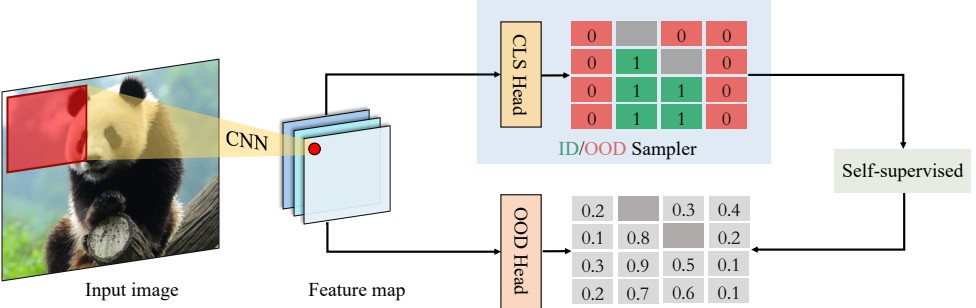

Figure 3: Self-supervised Sampling for Out-of-distribution Detection (SSOD). We adopt a self-supervised sampling scheme to train the OOD discrimination branch, *i.e.*, the OOD head with the supervised signals injected by the CLS head. The image patches in green/red/gray are ID/OOD/invalid signals identified by the CLS head. SSOD jointly trains these two branches.

## 2.3  SELF-SUPERVISED SAMPLING FOR OOD DETECTION

**Inspiration of SSOD.** In Fig. 1, we notice and verify that the image background is a good proxy for OOD data. Thus, we expect to extract the background information from the feature maps, which can be regarded as the natural OOD supervision. Prior studies, *e.g.*, Redmon et al. (2016) and Carion et al. (2020), have demonstrated that traditional neural networks are capable of retaining spatial information, *i.e.*, a position of the feature map reflects the corresponding position in the input image. With these foundations, we aim to design an OOD patch sampler to collect background information from the feature maps. In Fig. 2 (a), the ResNet-50 (He et al., 2016) trained on ImageNet (Russakovsky et al., 2015) downsamples the input image and yields a corresponding feature map ($\mathbb{R}^{H \times W \times C}$), where each image patch is projected to a feature vector ($\mathbb{R}^C$) located at the corresponding position. The classification head reports the category for each feature vector. We highlight the correctly classified patches in Fig. 2 (a) and (b). The results suggest that the confidence scores are much higher for patches contained in the main objects while lower for the backgrounds.

**Formulation of SSOD.** For a feature map within $\mathbb{R}^{C \times H \times W}$ (*i.e.*, the channel, height, and width) produced by the neural networks, we can apply the classifier along spatial axes and obtain a confidence score map within $\mathbb{R}^{M \times H \times W}$, where $M$ is the number of categories in ID data. The patches with a low confidence score, *e.g.*, lower than 5% on the ground-truth category, are recognized as OOD samples as highlighted in red in Fig. 2 (c) and (d). Symmetrically, an ID patch sampler selects some image patches with high confidence scores, *e.g.*, greater than 95%, as the ID samples (cf. Fig. 2, the green patches), helping to balance the positive (ID) and negative (OOD) samples.

Formally, we use $h(\cdot)$, $f_{cls}(\cdot)$, and $f_{ood}(\cdot)$ to denote the backbone removing the classification head, the multi-category classification head, and the binary ID/OOD discrimination head, respectively. Given an input image $x$ with label $y$, $X^{C \times H \times W} = h(x)$ is the feature map. The prediction result of the classification model is:

$$\hat{y} = f_{cls}(\text{GAP}(h(x))) = f_{cls}(\text{GAP}(X^{C \times H \times W})), \tag{11}$$

where GAP is the global average pooling over the spatial dimensions. Similarly, when applying $f_{cls}(\cdot)$ on each patch of $X^{C \times H \times W}$ without pooling operation, we can get the confidence score map $\hat{y}^{MHW} = f_{cls}(X^{C \times H \times W})$ within $\mathbb{R}^{M \times H \times W}$. Moreover, we pick the confidence along the target axis, *e.g.*, if the target label of $x$ is $j$, then we collect the confidence along the $j$-th axis of $M$, yielding a target confidence map within $\mathbb{R}^{H \times W}$, *i.e.*, $\hat{y}^{HW}$. We use the ID/OOD sampler to select patches with high/low scores on the target label as the ID/OOD supervision. Concretely, for $i \in \{1, 2, 3, ..., HW\}$, we obtain the following self-supervised OOD labels from the classification head:

$$y_i^{ood} = \begin{cases} 0, & \hat{y}_i^{HW} < 1 - \gamma \\ 1, & \hat{y}_i^{HW} \geq \gamma \\ \text{N/A}, & 1 - \gamma \leq \hat{y}_i^{HW} < \gamma \end{cases}, \tag{12}$$

where $\hat{y}^{HW}$ indicates the predicted confidence of each image patch belonging to the target category, and $\gamma$ is a confidence threshold, *e.g.*, 95%. Remind that the image patches assigned with the positive

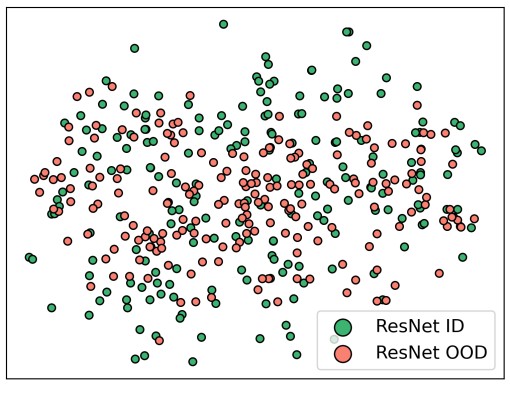 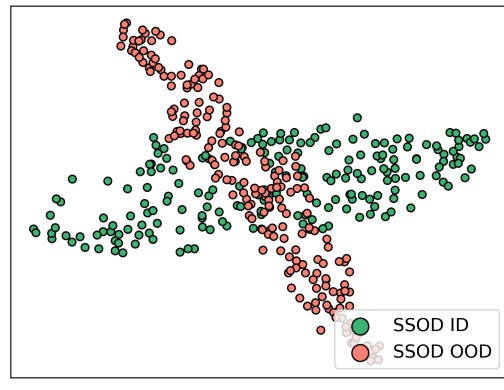

(a) ID *vs.* OOD features in ResNet-50.    (b) ID *vs.* OOD features in SSOD.

Figure 4: The t-SNE visualizations of deep features for the ImageNet (ID) and iNaturalist (OOD). **(a)**: features extracted by the conventional ResNet-50 (He et al., 2016), and the FPR95 is **54.99%**. **(b)**: features extracted by our SSOD, and the FPR95 is **14.80%**. The green/red dots represent ID/OOD features, *i.e.*, the ImageNet/iNaturalist. SSOD improves the ID/OOD discriminability notably. The overlap of ID/OOD features reflects the false positive rate (FPR95).

label are highlighted as green in Fig. 2, and the negative patches are marked in red. We drop the left image patches (*i.e.*, the non-highlighted patches in Fig. 2, the invalid patches in Fig. 3, and N/A in Eq 12), and therefore, they provide no ID/OOD supervisions during the training. With the OOD head, we obtain the ID/OOD prediction ($\hat{y}^{ood}$):

$$\hat{y}^{ood} = f_{ood}(X^{C \times H \times W}). \tag{13}$$

Since only a part of the image blocks is selected as ID/OOD supervision in Eq 12, consequently, the loss is performed on the corresponding predicted results in Eq 12 and Eq 13. The overall objective of SSOD is formulated with the cross entropy loss (CE):

$$\mathcal{L} = \text{CE}(\hat{y}, y) + \alpha\text{CE}(\hat{y}^{ood}, y^{ood}), \tag{14}$$

where $\alpha$ is a balance parameter, $\hat{y}^{ood} \in \mathbb{R}^{2 \times H \times W}$, and $y^{ood} \in \mathbb{R}^{H \times W}$. During the training/inference phase, the OOD factor of input images can be calculated as follows:

$$P(x \in \mathcal{S}_{\mathbb{ID}}|x) = \text{Sigmoid}(f_{ood}(\text{GAP}(X^{C \times H \times W}))), \tag{15}$$

where $\text{Sigmoid}$ function is used to predict the probability of input image belonging to the ID data. With the proposed SSOD above, we can train the OOD detection branch end-to-end with natural OOD supervisions sampled from the patches of ID background as illustrated in Fig. 3.

## 3 EXPERIMENTS

This section addresses the following problems: 1) How does SSOD perform on OOD detection benchmarks? 2) Whether SSOD is stable under different hyper-parameter settings? 3) Whether SSOD is generalizable across different backbones?

### 3.1 EXPERIMENTAL SETUP

**Benchmarks.** We employ large-scale benchmarks in OOD detection, including ImageNet (Russakovsky et al., 2015) groups, CIFAR-10 (Krizhevsky et al., 2009) groups, and hard OOD groups. In ImageNet groups, we set four OOD datasets, which are iNaturalist (Horn et al., 2018), SUN (Xiao et al., 2010), Places (Zhou et al., 2018), and Texture (Cimpoi et al., 2014). In CIFAR-10 groups, we set five OOD datasets, which are SVHN (Netzer et al., 2011), LSUN (Yu et al., 2015), iSUN (Xu et al., 2015), Texture (Cimpoi et al., 2014), and Places (Zhou et al., 2018). Under hard OOD setting, the ImageNet is employed as ID data while ImageNet-O (Huang & Li, 2021) and OpenImage-O (Wang et al., 2022) are selected as OOD data. The detailed information of these datasets and training/evaluation protocol is attached in Appendix A.3 and A.4.

Table 1: OOD detection results on ImageNet. ↓ **indicates lower is better, ↑ means greater is better.** We highlight the best and second results using bold and underlining. We use (w/) and (w/o) to indicate using and without using supervised contrastive learning. If not specified, all methods employ ResNet-50 pre-trained on ImageNet-1k as the backbone. F and R indicate FPR95 and AUROC.

| Method | iNaturalist | | SUN | | Places | | Texture | | Average | |
|---|---|---|---|---|---|---|---|---|---|---|
| | ↓ F | ↑ A | ↓ F | ↑ A | ↓ F | ↑ A | ↓ F | ↑ A | ↓ F | ↑ A |
| MSP | 54.99 | 87.74 | 70.83 | 80.86 | 73.99 | 79.76 | 68.00 | 79.61 | 66.95 | 81.99 |
| MSP (CLIP-B) | 40.89 | 88.63 | 65.81 | 81.24 | 67.90 | 80.14 | 64.96 | 78.16 | 59.89 | 82.04 |
| MSP (CLIP-L) | 34.54 | 92.62 | 61.18 | 83.68 | 59.86 | 84.10 | 59.27 | 82.31 | 53.71 | 85.68 |
| MaDist | 97.00 | 52.65 | 98.50 | 42.41 | 98.40 | 41.79 | 55.80 | 85.01 | 87.43 | 55.47 |
| ODIN | 47.66 | 89.66 | 60.15 | 84.59 | 67.89 | 81.78 | 50.23 | 85.62 | 56.48 | 85.41 |
| GODIN | 61.91 | 85.40 | 60.83 | 85.60 | 63.70 | 83.81 | 77.85 | 73.27 | 70.43 | 82.02 |
| KLM | 27.36 | 93.00 | 67.52 | 78.72 | 72.61 | 76.49 | 49.70 | 87.07 | 54.30 | 83.82 |
| Energy | 55.72 | 89.95 | 59.26 | 85.89 | 64.92 | 82.86 | 53.72 | 85.99 | 58.41 | 86.17 |
| KNN (w/o) | 59.08 | 86.20 | 69.53 | 80.10 | 77.09 | 74.87 | **11.56** | **97.18** | 54.32 | 84.59 |
| KNN (w/) | 30.18 | 94.89 | 48.99 | 88.63 | 59.15 | 84.71 | 16.97 | 94.45 | 38.82 | 90.67 |
| MOS | **9.28** | **98.15** | 40.63 | 92.01 | 49.54 | 89.06 | 60.43 | 81.23 | 39.97 | 90.11 |
| Fort (ViT-B) | 15.07 | 96.64 | 54.12 | 86.37 | 57.99 | 85.24 | 53.32 | 84.77 | 45.12 | 88.25 |
| Fort (ViT-L) | 15.74 | 96.51 | 52.34 | 87.32 | 55.14 | 86.48 | 51.38 | 85.54 | 43.65 | 88.96 |
| MCM (CLIP-B) | 30.91 | 94.61 | 37.59 | 92.57 | 44.69 | 89.77 | 57.77 | 86.11 | 42.74 | 90.77 |
| MCM (CLIP-L) | 28.38 | 94.95 | 29.00 | 94.14 | **35.42** | **92.00** | 59.88 | 84.88 | 38.17 | 91.49 |
| **SSOD (Ours)** | 14.80 | 96.91 | **28.52** | **94.33** | 38.92 | 90.78 | 45.32 | 87.02 | **31.89** | **92.26** |

**Comparable methods.** We choose both the classic and latest OOD detection methods for comparison. With regard to the classic schemes, we select the MSP (Hendrycks, 2017), MaDist (Lee et al., 2018b), ODIN (Liang et al., 2018), KL Matching (KLM) (Hendrycks et al., 2019), MaxLogit (MaxL) (Hendrycks et al., 2019), GODIN (Hsu et al., 2020), CSI (Tack et al., 2020), and MOS (Huang & Li, 2021). Besides, we also use the Energy (Liu et al., 2020), which is the representative of the score-based calibration method, and the KNN (Sun et al., 2022), which is one of the latest schemes, as our comparable methods. ResNet-18 and ResNet-50 (He et al., 2016) are chosen as the backbone for CIFAR-10 and ImageNet. KNN (Sun et al., 2022) has two different versions, *i.e.*, w/ and w/o contrastive learning. For a fair comparison with other methods, we employ **no** contrastive learning for all comparable techniques. Except for the above traditional methods, we also provide the results of large vision-language models, such as MCM (Ming et al., 2022), FLYP (Goyal et al., 2022), and Fort (Fort et al., 2021). The detailed information about them is attached in Appendix A.2.

## 3.2 COMPARISON WITH STATE-OF-THE-ARTS

**OOD detection on ImageNet.** We use iNaturalist, SUN, Places, and Texture as the OOD data. Following Sun et al. (2022), we randomly select 10,000 OOD images from each dataset for evaluation and keep the quantity of ID and OOD data the same for reliable FPR95. The KNN (Sun et al., 2022) methods have two versions, *i.e.*, using and w/o using contrastive learning. We provide both results for comparison and mark the contrastive version with (w/). MSP, Fort, and MCM employ strong backbones to improve the detection performance, such as CLIP (Radford et al., 2021) and ViT (Dosovitskiy et al., 2021). We annotate these backbones in the wake of each method. If not specified, all left methods use **no** contrastive learning and employ ResNet-50 as the backbone. We evaluate the performance of each method based on its averaged FPR95 and AUROC on the above four datasets. The technique yielding the best detection performance is highlighted using boldface, while the closely following one is underlined. From the results depicted in Table 1, we notice that SSOD yields competitive state-of-the-art performance on iNaturalist, SUN, and Places. Specifically, on iNaturalist, SSOD and MOS report comparable detection FPR95 as 14.80% and 9.28%. In contrast, other complicated methods such as ViT (Dosovitskiy et al., 2021) based Fort and CLIP (Radford et al., 2021) based MCM only achieve a FPR95 of 15.74% and 28.38%. Moreover, on SUN and Places, SSOD consistently establishes the top detection results (FPR95) as 28.52% and 38.92%, outperforming other techniques with a simple ResNet-50 backbone. On Texture, most methods

Table 2: OOD detection results on CIFAR-10. ↓ **indicates lower is better, ↑ means greater is better.** We highlight the best and second results using bold and underlining. All values are percentages.

| OOD | Metrics | Methods | | | | | | | |
|---|---|---|---|---|---|---|---|---|---|
| | | MSP | MaDist | ODIN | GODIN | Energy | CSI | KNN | SSOD |
| *SVHN* | ↓FPR95 | 59.66 | 9.24 | 20.93 | 15.51 | 54.41 | 37.38 | 24.53 | **2.12** |
| | ↑AUROC | 91.25 | 97.80 | 95.55 | 96.60 | 91.22 | 94.69 | 95.96 | **99.44** |
| *LSUN* | ↓FPR95 | 45.21 | 67.73 | 7.26 | 4.90 | 10.19 | 5.88 | 25.29 | **4.42** |
| | ↑AUROC | 93.80 | 73.61 | 98.53 | 99.07 | 98.05 | 98.86 | 95.69 | **99.11** |
| *iSUN* | ↓FPR95 | 54.57 | **6.02** | 33.17 | 34.03 | 27.52 | 10.36 | 25.55 | 10.06 |
| | ↑AUROC | 92.12 | **98.63** | 94.65 | 94.94 | 95.59 | 98.01 | 95.26 | 98.16 |
| *Texture* | ↓FPR95 | 66.45 | 23.21 | 56.40 | 46.91 | 55.23 | 28.85 | 27.57 | **1.91** |
| | ↑AUROC | 88.50 | 92.91 | 86.21 | 89.69 | 89.37 | 94.87 | 94.71 | **99.59** |
| *Places* | ↓FPR95 | 62.46 | 83.50 | 63.04 | 62.63 | 42.77 | 38.31 | 50.90 | **7.44** |
| | ↑AUROC | 88.64 | 83.50 | 86.57 | 87.31 | 91.02 | 93.04 | 89.14 | **98.42** |
| *Average* | ↓FPR95 | 57.67 | 37.94 | 36.16 | 32.80 | 38.02 | 24.20 | 30.80 | **5.19** |
| | ↑AUROC | 90.90 | 89.29 | 92.30 | 93.52 | 93.05 | 95.90 | 94.15 | **98.94** |
| | ↑ID ACC | 94.21 | 94.21 | 94.21 | 93.96 | 94.21 | **94.38** | 94.21 | 94.17 |

fail to identify the OOD images and demonstrate high FPR95. Nevertheless, SSOD detects most outliers of Texture and reports a FPR95 of 45.32%, which is better than all methods except for KNN (Sun et al., 2022). This failure case is caused by the overlap between ImageNet-1k and Texture (cf. Appendix A.5). Overall, SSOD obtains the FPR95 and AUROC as 31.89% and 92.26% on the four above large-scale datasets, surpassing the closely following method with **-6.28%** FPR95 and **+0.77%** AUROC. Besides, we also demonstrate the confidence distribution of ID/OOD images in Appendix Fig. 6, evidencing that SSOD significantly reduces the overlap between ID and OOD confidence distributions, promising a much better performance in detecting OOD examples.

**OOD detection on CIFAR-10.** Following previous methods, we adopt SVHN, LSUN, iSUN, Texture, and Places as the OOD data. All comparable techniques take ResNet-18 as the backbone. SSOD resorts to no pre-trained weights and trains the classifier from scratch. CIFAR-10 and other OOD samples are resized to $224 \times 224$ to obtain sufficient background information from the images. The detailed results are depicted in Table 2. On iSUN, the detection performance of SSOD is marginally lower than MaDist, while on the left four datasets, SSOD consistently outperforms other methods by a significant margin. In particular, on Places and Texture, SSOD reduces the FPR95 of **30.87%** and **21.30%** compared to the closely following one, evidencing its effectiveness. From an overall view, on the above OOD datasets, SSOD improves the detection ability of ResNet-18 by **-19.01%** FPR95 and **+3.04%** AUROC on average, which can be established as one of the state-of-the-arts.

**Hard OOD detection.** ImageNet-O (Huang & Li, 2021) and OpenImage-O (Wang et al., 2022) are employed as hard OOD examples for ImageNet, since they have similar natural scenes to those in ImageNet. All methods use ResNet-50d (He et al., 2016), a variant of ResNet-50, as the backbone. As shown in Table 3, SSOD achieves top-ranked performance on OpenImage-O (Wang et al., 2022). Since the ImageNet-O mainly contains adversarial images, leading to the classifier's wrong prediction, SSOD reports higher FPR95 compared to the best previous methods. Overall, SSOD still achieves comparable state-of-the-art performance, yielding 4.38% higher FPR95 compared to ViM (Wang et al., 2022) and 1.15% lower AUROC compared to ReAct (Sun et al., 2021).

## 3.3 ABLATION STUDY

**Ablations on hyper-parameter $\alpha$.** Recall that $\alpha$ controls the importance of loss generated by the OOD head, balancing classifiers' classification performance and OOD detection ability. We employ CIFAR-10 (Krizhevsky et al., 2009) and Places (Zhou et al., 2018) as the ID and OOD data to validate the stability of $\alpha$. SSOD uses ResNet-18 as the backbone. From the ablations depicted in Table 4, we notice that the classifier detects OOD input better with increasing $\alpha$, while the ID ACC is gradually descending. We expect to boost the robustness of classifiers while not affecting the model's performance. Therefore, we set $\alpha$ to 1.0 throughout the experiments.

Table 3: Hard OOD detection results.

| Method | ImageNet-O | | OpenImage-O | | Average | |
|---|---|---|---|---|---|---|
| | ↓F | ↑A | ↓F | ↑A | ↓F | ↑A |
| MSP | 93.85 | 56.13 | 63.53 | 84.50 | 78.69 | 70.32 |
| Energy | 90.10 | 53.95 | 76.83 | 75.95 | 83.47 | 64.95 |
| ODIN | 93.25 | 52.87 | 64.49 | 81.53 | 78.87 | 67.20 |
| MaxL | 92.65 | 54.39 | 65.50 | 81.50 | 79.08 | 67.95 |
| KLM | 88.50 | 67.00 | 60.58 | 87.31 | 74.54 | 77.16 |
| ReAct | **72.85** | **81.15** | 60.79 | 85.30 | _66.82_ | **83.23** |
| MaDist | 78.45 | 68.02 | 55.91 | 89.52 | 67.18 | 78.77 |
| ViM | _76.00_ | _74.80_ | **50.45** | **90.76** | **63.23** | _82.78_ |
| **SSOD** | 79.80 | 74.43 | _55.41_ | _89.73_ | 67.61 | 82.08 |

Table 4: Ablations on the hyper-parameter $\alpha$.

| $\alpha$ | 0.5 | 0.7 | 1.0 | 1.2 | 1.5 |
|---|---|---|---|---|---|
| ↓FPR95 | 11.64 | 9.44 | 7.44 | 8.56 | 10.80 |
| ↑AUROC | 97.33 | 97.74 | 98.42 | 98.19 | 97.49 |
| ↑ID ACC | 94.25 | 94.17 | 94.17 | 92.53 | 91.36 |

Table 5: Ablations of balance schemes.

| Scheme | LW | DR | LWB |
|---|---|---|---|
| ↓FPR95 | 10.72 | 8.48 | 7.44 |
| ↑AUROC | 97.66 | 98.05 | 98.42 |
| ↑ID ACC | 93.97 | 94.11 | 94.17 |

Table 6: OOD detection with different backbones. We highlight the best and second results using bold and underlining. ♦ indicates the SSOD counterpart.

| Method | iNaturalist | | Places | | Average | |
|---|---|---|---|---|---|---|
| | ↓F | ↑A | ↓F | ↑A | ↓F | ↑A |
| R-18 | 60.00 | 86.32 | 76.96 | 76.86 | 67.42 | 81.65 |
| R-18♦ | 19.96 | 95.21 | 46.48 | 88.31 | **33.22** | **91.76** |
| R-34 | 57.88 | 86.44 | 75.96 | 78.16 | 66.92 | 82.30 |
| R-34♦ | 15.88 | 96.28 | 43.44 | 87.33 | **29.66** | **91.81** |
| R-50 | 54.99 | 87.74 | 73.99 | 79.76 | 64.49 | 83.75 |
| R-50♦ | 14.80 | 96.91 | 38.92 | 90.78 | **26.86** | **93.85** |
| R-101 | 56.58 | 86.08 | 70.78 | 80.18 | 63.68 | 83.13 |
| R-101♦ | 18.78 | 95.56 | 42.72 | 87.84 | **30.75** | **91.70** |
| D-121 | 59.44 | 86.79 | 69.88 | 80.12 | 64.66 | 83.46 |
| D-121♦ | 21.64 | 94.84 | 44.88 | 85.51 | **33.26** | **90.18** |
| RegN | 54.96 | 87.74 | 70.28 | 80.03 | 62.62 | 83.89 |
| RegN♦ | 16.36 | 96.05 | 35.80 | 89.79 | **26.08** | **92.92** |
| MobN | 52.52 | 88.19 | 73.36 | 79.41 | 62.94 | 83.80 |
| MobN♦ | 24.76 | 94.47 | 42.52 | 88.38 | **33.64** | **91.43** |

**Imbalance issue between ID/OOD features.** During the training of the OOD head, we obtain much more background features since the objects only occupy a small part of the image. To promote training stability, we design three ways to tackle this issue, which are *Loss Weighting* (LW), *Data Resampling* (DR), and *Loss-Wise Balance* (LWB). LW multiplies a balance factor on the loss generated by the background features, DR randomly samples equivalent ID/OOD features within each image, and LWB calculates the cross entropy generated by the ID/OOD features separately and picks their mean value as the loss objective. CIFAR-10 and Places are the ID and OOD data. Based on the ablations depicted in Table 5, SSOD employs LWB for data balancing.

**OOD detection across different model architectures.** To validate the transfer ability of SSOD on different model architectures, we select ImageNet as the ID data, iNaturalist, and Places as the OOD data. Considering the deployment on portable devices, we test both the conventional and lite models, such as ResNet (He et al., 2016) series, DenseNet-121 (Huang et al., 2017), RegNet (Y-800MF) (Radosavovic et al., 2020), and MobileNet (V3-Large) (Howard et al., 2019). Table 6 uses ↓F and ↑A to indicate the FPR95 and AUROC. R-18/34/50/101 are ResNet-18/34/50/101. D-121, RegN, and MobN are DenseNet-121, RegNet, and MobileNet. Compared to Table 1, methods (♦) shown in Table 6 achieve state-of-the-art performance, evidencing the scalability of SSOD. The improvements between the vanilla classifier and its improved SSOD version (♦) are also significant, *e.g.*, **-37.63%** FPR95 on ResNet and **-36.54%** FPR95 on RegNet.

## 4 CONCLUSIONS

This paper proposes a probabilistic framework that divides the robust classification into ID and OOD factors. This provides a comprehensive overview of existing OOD methods and highlights the critical constraint of relying on pre-trained features. To address this limitation, we introduce an end-to-end scheme called SSOD, which optimizes the OOD detection jointly with the ID classification. This approach leverages OOD supervision from the background information of ID images, eliminating the data-collecting need. Extensive experiments have validated that SSOD achieves competitive performance in detecting OOD data, which can be a starting point for future research.

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

## A  APPENDIX

### A.1  RELATED WORK

**Score-based posterior calibration.** This line of research aims to find differences between the ID and OOD data, thus designing model-specific discriminative functions to identify the OOD samples. The related work includes ODIN (Liang et al., 2018), LogitNorm (Wei et al., 2022), GradNorm (Huang et al., 2021), ReAct (Sun et al., 2021), Energy (Liu et al., 2020), and CIDER (Ming et al., 2023), to name a few. Generally, these methods are usually pre- or post-processing schemes that demonstrate no need for retraining the neural networks. Although these methods above report considerable performance improvements and sometimes are training efficient, they do not necessarily lead to significant generalization ability. For example, ReAct (Sun et al., 2021) investigates the distinct behaviors of ID and OOD data after ReLU function, and therefore, it fails to perform on architectures adopting other activations, such as GELU, Sigmoid, and Tanh, *etc*. Similarly, the ODIN (Liang et al., 2018) investigates post-processing schemes specially designed for Softmax, *i.e.*, the temperature scaling. These specific designs promote OOD detection but limit the model's scalability. In contrast, our SSOD doesn't suffer from this limitation as it addresses the OOD detection directed by Bayes' theorem, which widely holds in general scenarios.

**Auxiliary supervision from synthetic OOD data.** The lack of OOD supervision is a critical factor leading to unsatisfactory performance in OOD detection. Thus, significant interest has been raised in generating synthetic OOD data. Existing approaches tackling this issue can be roughly divided into two manners, which are feature and image generation. The former samples OOD features from the ID boundary, such as VOS (Du et al., 2022), or generates them using GAN, such as BAL (Pei et al., 2022). In contrast, the image generation yields more training tax since it directly generates the OOD images, such as Conf (Lee et al., 2018a), SBO (Möller et al., 2021), MG-GAN (Dendorfer et al., 2021), NAS-OOD (Bai et al., 2021), CODEs (Tang et al., 2021), and VITA (Chen et al., 2022b). In summary, these methods either introduce bias as they only consider the approximated feature space or are costly due to the image generation. Unlike the above methods, SSOD avoids feature bias and training tax by utilizing the universal local property of neural networks, extracting realistic OOD supervision from the ID images without generation cost.

### A.2  OOD DETECTION VIA VISION-LANGUAGE PRE-TRAINING.

Recently, vision-language pre-training and large language models (LLMs) have become a trend to include various downstream tasks, such as text-image retrieval (Radford et al., 2021), VQA (Chen et al., 2022a), objects segmentation (Kirillov et al., 2023), and image captioning (Li et al., 2023), to name a few. Empowered by the massive amount of training data and model parameters, we expect to see blossom in detecting OOD data with these cross-modal models. For instance, MCM (Ming et al., 2022) and Fort (Fort et al., 2021) resort to the two-tower architectures, *i.e.*, CLIP (Radford et al., 2021), for matching the text descriptions within a closed set and open images. The matching score is used for ruling out the OOD data. Except for designing matching rules along the CLIP paradigm, FLYP (Goyal et al., 2022) notices that vison-language fine-tuning also plays a decisive role in the final OOD detection performance.

### A.3  BENCHMARKS

We perform experiments on ImageNet (Russakovsky et al., 2015) and CIFAR-10 (Krizhevsky et al., 2009). For ImageNet, we follow the settings from Sun et al. (2022) and employ ImageNet-O (Huang & Li, 2021), OpenImage-O (Wang et al., 2022), iNaturalist (Horn et al., 2018), SUN (Xiao et al., 2010), Places (Zhou et al., 2018), and Texture (Cimpoi et al., 2014) as the OOD images. For CIFAR-10, we select SVHN (Netzer et al., 2011), LSUN (Yu et al., 2015), iSUN (Xu et al., 2015), Places (Zhou et al., 2018), and Texture (Cimpoi et al., 2014) as the OOD images. Images in CIFAR-10 and ImageNet are resized and cropped to $224 \times 224$. All OOD images enjoy the identical pre-processing method. The detailed information of all employed datasets is presented as follows.

**ImageNet.** ImageNet (Russakovsky et al., 2015) is well known in image classification problems, containing 1,000 classes from the natural scene such as *tiger*, *goldfish*, and *house*, to name a few. This dataset is used as the ID data, expecting to get higher confidence from the classifier, *i.e.*, higher OOD factor (cf. Eq 7).

**CIFAR-10.** CIFAR-10 (Krizhevsky et al., 2009) is smaller compared to the ImageNet. It consists of tiny images within 10 classes, such as *airplane*, *bird*, *dog*, *etc*. All images contained in CIFAR-10 are in the shape of $32 \times 32$. Our experiments treat CIFAR-10 as the in-distribution data and resize images into $224 \times 224$ to get bigger feature maps. The above operation marginally influences the classification performance since up-sampling these tiny images yields no information gain.

**SVHN.** This dataset (Netzer et al., 2011) indicates the street view house number, consisting of digit numbers from the natural street view. Following Sun et al. (2022), we randomly select 10,000 images from this dataset to serve as the OOD data. All pictures from OOD data are expected to get lower confidence from the classifier.

**LSUN.** This dataset is used for visual recognition, which was presented in Yu et al. (2015). It consists of over one million labeled images, including 10 scene and 20 object categories. Following Sun et al. (2022), 10,000 images from LSUN are treated as the OOD data.

**iNaturalist.** The existing dataset in the image classification problem usually has a uniform distribution across different objects and categories. However, in the real world, the images could be heavily imbalanced. To bridge this gap between experimental and practical settings, iNaturalist (Horn et al., 2018), consisting of over 859,000 images within about 5,000 species (*i.e.*, planets and animals), is presented. We randomly selected 10,000 images from this dataset as OOD pictures.

**iSUN.** This dataset is constructed based on the SUN (Xiao et al., 2010). iSUN (Xu et al., 2015) is a standard dataset for scene understanding, containing over 20,000 images from SUN database. We use 10,000 iSUN images as the OOD data.

**Texture.** This dataset consists of images carrying vital characters, *i.e.*, patterns and textures of natural objects. Presented in Cimpoi et al. (2014), Texture aims at supporting the analytical dimension in image understanding. In our experiments, we treat Texture as the OOD data. Data cleaning has been performed in Sun et al. (2022).

**Places.** This dataset is used for scene recognition, which was presented by Zhou et al. (2018), including over 10 million images such as *badlands*, *bamboo forest*, and *canal*, *etc*. Following Sun et al. (2022), we use 10,000 of these images to play the role of OOD data.

**ImageNet-O.** This dataset is released in MOS (Huang & Li, 2021), which is built as hard adversarial OOD data for ImageNet, consisting of 2,000 images.

**OpenImage-O.** This dataset is presented in ViM (Wang et al., 2022), which follows natural class statistics and is manually labeled at the image level. The whole dataset contains 17,632 images.

## A.4 TRAINING AND EVALUATION.

All images used in our experiments are resized to $224 \times 224$. We use AdamW as the optimizer. The learning rate starts from 1e-4 and halves every 30 epochs. The experiment runs on 8 NVIDIA Telsa V100 GPUs. The batch size is set to 256, *i.e.*, $32 \times 8$, each GPU is allocated with 32 images. We store the checkpoints yielding the best FPR95 performance. About the evaluation, we report the false positive rate of the OOD dataset when the true positive rate of ID images is 95%, *i.e.*, FPR95. We also compare the area under the receiver operating characteristic curve (AUROC) and the classification accuracy of ID images (ID ACC). We keep the quantity of ID and OOD data consistent following Hendrycks (2017).

## A.5 FALIURE CASE ANALYSES

Recall that SSOD extracts OOD information from the background of training images and employs them as the proxy of OOD characters, revealing the potential of suffering limited diversity of the OOD supervision if the training images are not diverse. This phenomenon will be perceived if the OOD data differs in domains from the training images. For example, the training images are natural scenes, while the testing OOD data is synthetic color blocks or textures. To check this issue, We train the SSOD on ImageNet (Russakovsky et al., 2015) while testing it on Texture (Cimpoi et al., 2014). From the results depicted in Table 1, though SSOD achieves top-ranked performance, it is worse than KNN (Sun et al., 2022), increasing the FPR95 by about **33.76%**. The overlap between ImageNet and Textures causes this issue. Concretely, many images in the Texture (Cimpoi et al., 2014) carry

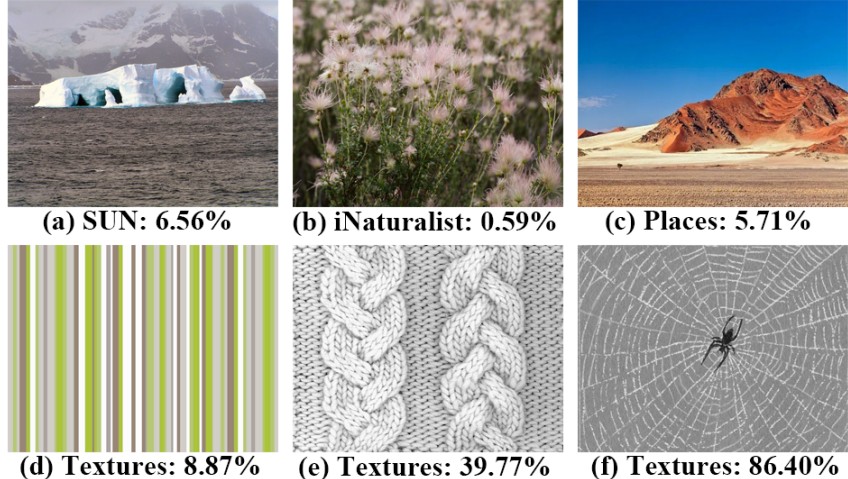

Figure 5: Confidence of OOD images assigned by SSOD. **Successful cases**: (a), (b), (c), and (d). These images are randomly sampled from several OOD datasets and assigned nominal ID confidence by the SSOD. **Faliure cases**: (e) and (f). These images carry vital symbols of objects appearing in ImageNet, *e.g.*, (e) is the *braided* and (f) is *cobwebbed*, which are similar with the *knot* and *spider* in ImageNet. This phenomenon indicates the importance of data cleaning during the evaluation phase.

vital symbols of objects included in ImageNet (cf. Fig. 5). These overlaps lead to the inefficiency of SSOD during the comparison with KNN (Sun et al., 2022) on Texture dataset in Table 1.

## A.6 VISUALIZATION RESULTS

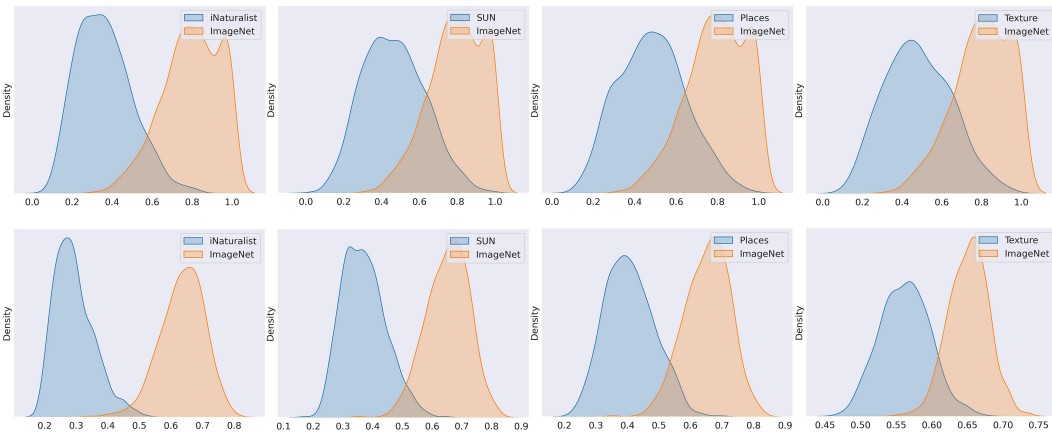

Figure 6: Confidence distribution of images from ImageNet (ID) and other OOD datasets. The curve in orange indicates the confidence distribution of ImageNet, while the curves in blue are that of OOD images, including iNaturalist, SUN, Places, and Texture from left to right. In the figures above, the $x$-label and $y$-label are the confidence of images belonging to ID data and their overall density. **Top**: MSP yields a bigger overlap between the ID and OOD confidence distribution, which means a higher FPR95. **Bottom**: SSOD assigns higher confidence to ID images, and the overlap between ID/OOD confidence is relatively small, corresponding to the low FPR95.

In the figure above, the curves in orange are the confidence distribution of pictures in ImageNet, while the blue curves are that of OOD data, including iNaturalist, SUN, Places, and Texture. As clearly depicted in Fig. 6, MSP (top) yields greater overlap between the two distributions, which means more ID/OOD images are confused. In contrast, SSOD (bottom) significantly reduces the overlap between ID and OOD confidence distributions, promising a much better performance in detecting OOD examples.

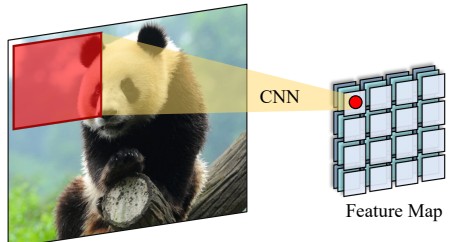

(a) The local receptive field of convolution.

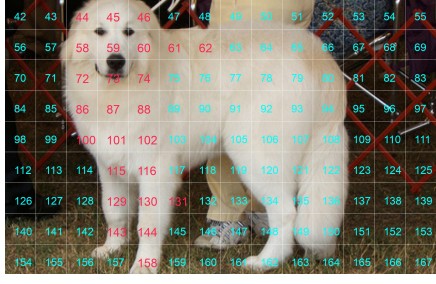

(b) Patch-wise classification results using ResNet-50.

Figure 7: Illustration of the spatial correspondence between the image and its feature. **(a)**: Convolutional neural networks project the image patch to a feature point within the feature map. The pooled feature map is then fed into the linear classifier to determine its category. The representation scope (*i.e.*, receptive field) of each feature point is dominated by the model's down-sampling rate. **(b)**: The trained classification head of ResNet-50 (He et al., 2016) can correctly identify the feature point carrying significant characters. Concretely, we feed the model with a *Great Pyrenees* image, and the outputted feature map is directly used for classification without pooling. We mark the correctly predicted image patch as red and otherwise lime green. It is clear that the neural network can correctly identify the determinative parts within the image. The number in the right panel indicates the index of the image patch, and we only demonstrate a part of the image for saving space.

