# OpenReview forum: "Image Background Serves as Good Proxy for Out-of-distribution Data"
_ICLR.cc/2024/Conference — ICLR 2024 poster_

### Official Review · Reviewer_EtXA · 2023-10-26

**Soundness:** 2 fair
**Presentation:** 2 fair
**Contribution:** 2 fair
**Rating:** 3
**Confidence:** 4

**Summary:**

The text addresses challenges in Out-of-distribution (OOD) detection, emphasizing the need for a unified perspective and the generation of compact boundaries between in-distribution (ID) and OOD data without explicit OOD samples. A general probabilistic framework is proposed to interpret various existing OOD detection methods, providing insights for future research. Concurrently, a Self-Supervised Sampling for OOD Detection (SSOD) model is introduced, which leverages natural OOD signals from ID data through convolution’s local property, allowing for the joint optimization of OOD detection and ID classification in an end-to-end manner. Extensive experiments demonstrate that SSOD achieves state-of-the-art performance on large-scale benchmarks, significantly outperforming previous methods, and showing remarkable results on both standard and challenging OOD datasets.

**Strengths:**

1. The paper is well-written and easy to understand.
2. The author conducts extensive experiments for evaluation.
3. The visualization is helpful to better understand SSOD.

**Weaknesses:**

1. SSD [1] also uses a self-supervised algorithm for OOD detection; however, the article does not make any comparisons with SSD.
2. As shown in Tables 2, 4, and 5, SSOD results in a decrease in the model's accuracy for ID classification, which is not permissible for OOD detection. OOD detection requires the model to identify OOD data without affecting the accuracy of ID classification.
3. The results reported in Table 1 show that SSOD does not achieve state-of-the-art (SOTA) performance on all OOD datasets, and there are many works with better performance not compared in Table 1: React [2], Dice [3], Ash [4].
4. The motivation is not convincing enough: I do not agree with the author's critique of the two-stage manner approach. Post-processing algorithms are actually more suitable for adapting to various pre-trained models, regardless of whether the model is trained with supervised learning or self-supervised learning.
5. Table three indicates that SSOD performs poorly on hard OOD detection tasks, does this highlight a flaw in the algorithm: its inability to differentiate between ID and OOD with similar backgrounds? Take the following more realistic example of OOD input: if a network trained to distinguish between different types of apples (e.g., fuji, red, honey-crisp, etc.) is presented with a different fruit as OOD data (e.g., peach, plum, tomato, etc.), the background features could be similar for both ID and OOD data. Alternatively, consider a scenario like scene classification (indoor scene vs beach scene vs forest scene, just as a hypothetical example). In this case, there is no background as such, because the entire scene constitutes the 'foreground.' I am very curious about how SSOD would perform in such scenarios.
6. I recommend the author conduct experiments about ViT architectures.

[1]  Sehwag, Vikash, Mung Chiang, and Prateek Mittal. "Ssd: A unified framework for self-supervised outlier detection." ICLR 2021.

[2] Sun, Yiyou, Chuan Guo, and Yixuan Li. "React: Out-of-distribution detection with rectified activations." Advances in Neural Information Processing Systems 34 (2021): 144-157.

[3] Sun, Yiyou, and Yixuan Li. "Dice: Leveraging sparsification for out-of-distribution detection." European Conference on Computer Vision. Cham: Springer Nature Switzerland, 2022.

[4] Djurisic, Andrija, et al. "Extremely Simple Activation Shaping for Out-of-Distribution Detection." The Eleventh International Conference on Learning Representations. 2022.

**Questions:**

see Weakness

---

> ### Author Response · Authors · 2023-11-14
> **Response from the Authors**
>
> Thanks for your valuable suggestions. We detail our response below.
> - **Response to weakness 1**: We notice that the original manuscript of SSD only provides AUROC while ignoring the FPR95, which is more important in OOD detection. Nevertheless, we provide the comparison (AUROC) below. The CIFAR-10 is employed as the in-distribution data.
> | OOD data | SVHN | LSUN | iSUN | Texture | Places | Average |
> | --- | --- | --- | --- | --- | --- | --- |
> | SSD | 99.60 | 96.50 | None | 97.60 | 95.20 | 97.225 |
> | Ours (SSOD) | 99.44 | 99.11 | 98.16 | 99.59 | 98.42 | **98.944** |
> - **Response to weakness 2**: SSOD only introduces marginal influence to the ID ACC. As a researcher and a practical developer of online applications, we focus on the overall performance of the method instead of entangling in these imperceptible changes. We provide the influence that SSOD introduced on ID ACC below.
> | Method | ResNet-50 | SSOD (ResNet-50) | ResNet-101 | SSOD (ResNet-101) | MobileNet V3L | SSOD (MobileNet V3L) | ResNet-18 | SSOD (ResNet-18) |
> | --- | --- | --- | --- | --- | --- | --- | --- | --- |
> | Dataset | ImageNet-1K | ImageNet-1K | ImageNet-1K | ImageNet-1K | ImageNet-1K | ImageNet-1K | CIFAR-10 | CIFAR-10 |
> | ID ACC | 76.13 | 76.09 (-0.04%) | 77.37 | 76.92 (-0.45%) | 74.04 | 73.94 (-0.1%) | 94.21 | 94.17 (-0.04%) |
> - **Response to weakness 3**: We evaluate our proposed SSOD on several benchmarks as shown in Table 1 and Table 2. The averaged results suggest that SSOD surpasses other methods by a large margin, i.e., **-6.28%** FPR95 on ImageNet and **-19.01%** FPR95 on CIFAR-10. Moreover, as a researcher in this field and to the best of our knowledge, our team doesn't believe there exists a method that performs absolutely better than all other existing methods on all benchmarks. Besides, we compare over 13 (MSP, MaDist, ODIN, GODIN, KLM, Energy, KNN, MOS, Fort, MCM, CSI, ReAct, ViM, and their variants) methods in our experiments, which is quite sufficient and enough to represent the current research situation. ReAct is shown in `Table 3`. SSOD achieves comparable performance on hard OOD detection as ReAct. More comparisons are demonstrated below. The ImageNet-1K is set as ID data and we compare the FPR95 of ReAct, Dice, and Ash. We will also attach this part to our revised manuscript. Thanks for your suggestion in filling the gap appearing in our experiments. Honestly, based on the results demonstrated in the original manuscript of Ash, it performs better than most current methods, and our SSOD also achieves top-ranked performance among them. **The following results are FPR95, lower is better.**
> | Dataset (ID=ImageNet-1K) | iNaturalist | SUN | Places | Texture | Average |
> | --- | --- | --- | --- | --- | --- |
> | ReAct | 20.38 | 24.20 | 33.85 | 47.30 | 31.43 |
> | Dice | 25.63 | 35.15 | 46.49 | 31.72 | 34.75 |
> | Ash | 14.21 | 22.08 | 33.45 | 21.17 | 22.73 |
> | Ours (SSOD) | 14.80 | 28.52 | 38.92 | 45.32 | 31.89 |
>
>    | Dataset (ID=CIFAR-10) | SVHN | LSUN | iSUN | Texture | Places | Average |
>    | --- | --- | --- | --- | --- | --- | --- |
>    | ReAct | 41.64 | 11.46 | 12.72 | 43.58 | 43.31 | 30.54 |
>    | Dice | 25.99 | 3.91 | 4.36 | 41.90 | 48.59 | 24.95 |
>    | Ours (SSOD) | 2.12 | 4.42 | 10.06 | 1.91 | 7.44 | 5.19 |
> - **Response to weakness 4**: I think that there exist some misunderstandings. We agree with you that both post-processing and pre-processing methods are great for tackling OOD detection, such as ODIN, GODIN, ReAct, and KNN. However, when concerning practical applications, we find two obstacles that these methods encounter. Firstly, we have no suitable pre-trained weights for some specific tasks such as face recognition. This makes it costly to perform these methods. Besides, the deployment is usually complex as we have to compute and store many intermediate results or features. Secondly, our proposed SSOD can better utilize the image data from the target scenario since it's end-to-end trainable, which is more suitable in some situations than the two-stage manner approach. Again, we are here to emphasize that the two-stage manner methods are quite great for promoting the development of OOD detection. **The motivation of SSOD is to inject more insights into this research direction and provide a more suitable technical proposal in some specific scenarios.** For example, in surveillance video processing, if we have some data from the target scenarios, our SSOD can learn their backgrounds as OOD characters, and therefore, reduce false alarms when nothing appears on the video screen.

---

> > ### Author Response · Authors · 2023-11-14
> > **Response from the Authors**
> >
> > - **Response to weakness 5**: SSOD encounters failures if the CLS head performs poorly. **In image recognition tasks, there always exists some vital and discriminative characters for neural networks to identify them.** These vital characters will be employed as ID supervision, and the less important characters will serve as OOD supervision. For example, given the classification task as indoor vs. forest vs. beach, the model will capture important characters such as sofa, trees, and rivers as the ID characters while the person is the OOD character since they can appear in all scenes.
> > - **Response to weakness 6**: ViT is based on self-attention mechanisms. The methodology of SSOD is to extract pure background information from the input images. These prerequisites tell us that the key to applying SSOD to self-attention based models is to extract local image features. This topic has a lot of content and many details to discuss. Therefore, we will not include this part in this paper. **Our current solution is to sample image tokens before self-attention in ViT and add an OOD token to learn the discriminative characters of ID and OOD data.** The OOD token is similar to the CLS token, and we also add an OOD head to perform ID/OOD classification. Currently obtained results are better than ResNet on ImageNet, and we will explore more sampling methods. The manuscript of this extended version will be made publicly available on Arxiv after completion.

---

### Official Review · Reviewer_u8Y6 · 2023-10-28

**Soundness:** 1 poor
**Presentation:** 3 good
**Contribution:** 2 fair
**Rating:** 3
**Confidence:** 3

**Summary:**

The paper addresses challenges in Out-of-distribution (OOD) detection by:
1. Proposing a unified probabilistic framework to understand existing methods.
2. Introducing a new model, SSOD, that uses natural OOD proxy from in-distribution data without needing explicit OOD samples.
3. Demonstrating that SSOD significantly outperforms previous methods on major benchmarks.

**Strengths:**

1. The paper is articulately composed and organized, facilitating a clear understanding of most sections.
2. To my understanding, the method introduced is innovative.
3. The study is underpinned by a comprehensive set of experiments.

**Weaknesses:**

Major Points:

1. **Concerns about the First Contribution:**

a. The derivation in Eqs 1-6 appears not helpful to the proposed method and Eq. 7 could be introduced more directly with

$$P(w_i|x)=P(w_i,x\in S_{ID}|x) = P(w_i|x\in S_{ID},x) \cdot P(x\in S_{ID}|x), i=1,\dots,M$$

b. The analysis of prior methods seems not necessarily dependent on a probabilistic perspective, since $P(x\in S_{ID}|x)$ is essentially a rephrasing of existing methods (in A.1).

Thus, the emphasis on the probabilistic viewpoint might not be as novel as suggested.

2. **Impact of Confidence Threshold:** How does the confidence threshold, $\gamma$, in Eq. 9 influence the performance? Could the authors elaborate on how they determined its value during experiments?

3. **Bias in the Proposed Method:** The paper indicates that existing post-hoc methods are influenced by biases from pretrained models. Yet, as Table 3 reveals, the proposed technique doesn't perform optimally on ImageNet-O, which contains adversarial images for ImageNet. Given that the method's training relies on both the original dataset and model's intermediate predictions, is it possible the method still suffers from the bias?

4. **Effect of OOD Training Target on ID ACC:** In the experiments, what is the ID ACC performance when $\alpha=0$? Essentially, does including the OOD training target lead to a noticeable decrease in ID ACC?

5. **Fairness of Comparison in Experiments:** In the experimentation section, multiple methods such as MSP, ODIN, ReAct, and many others are post-hoc techniques that work off a fixed pre-trained model. These methods don't adjust the training process, distinguishing them from the proposed approach. As a result, juxtaposing these techniques might not offer a balanced comparison. The majority of experiments may not necessarily highlight the superiority of the proposed method. It would be beneficial for the authors to contrast their strategy with other training-based techniques.

Minor Points:
1. **Table Formatting Issues:** In Tables 4 & 5, the shaded regions seem to obscure the lines, affecting clarity.

2. **Discussion Location:** In the introduction, the authors state that "various OOD methods can be analyzed, with main differences and key limitations clearly identified". However, this discussion is relegated to the appendix. It would be more helpful if the main content were self-contained and inclusive of this analysis.

**Questions:**

Please see the content in Weakness.

---

> ### Author Response · Authors · 2023-11-13
> **Response from the Authors**
>
> Thanks for your valuable reviews. We detail our response below.
> - **Response to weakness 1 (a)**: The derivation from `Eq 1` to `Eq 6` is based on Bayes's theorem. Concretely, based on the total probability formula, we have $P(w_i|x)=P(w_i|x\in S_{ID}, x)P(x\in S_{ID}|x)+P(w_i|x\in S_{OOD}, x)P(x\in S_{OOD}|x)$. As illustrated in `Section 3.1`, $w_i$ indicates the ID categories, and $S_{ID}$ has no overlap with $S_{OOD}$. Therefore, the term $P(w_i|x\in S_{OOD}, x)$ equals zero. This makes `Eq 7` hold conditionally.
> - **Response to weakness 1 (b)**: The probabilistic framework is vital for robust classification. As we can see from `Eq 7`, robust classification consists of conventional multi-category classification (ID factor) and OOD detection (OOD factor). **This perspective provides a unified view of open world recognition.** OOD detection is an important part of robust classification, and we show their connection clearly.
> - **Response to weakness 2**: The ablations on $\gamma$ are provided in our response to **Reviewer #ajFn: Response to question 2**.
> - **Response to weakness 3**: Honestly, SSOD is quite effective but not perfect. The methodology of SSOD is to obtain OOD supervision from the CLS head. Therefore, if the CLS head fails to detect the target object of the ID dataset, its generated ID/OOD supervision is likely to be invalid. This tells us that SSOD suffers from the bias when the CLS head fails to identify the input images, i.e., fails to perform classification. We argue that it's hard to design a perfect method that can handle all scenarios. We present SSOD for injecting new thoughts into the sub-field of OOD syntheses.
> - **Response to weakness 4**: If $\alpha$ is set to zero, SSOD degenerates into conventional classification, and the ID ACC equals that of the pre-trained models on ImageNet-1K. Marginal influence is introduced by SSOD on ID ACC, and we demonstrate the detailed results below. All pre-trained weights are released by Pytorch officially.
> | Method | ResNet-50 | SSOD (ResNet-50) | ResNet-101 | SSOD (ResNet-101) | MobileNet V3L | SSOD (MobileNet V3L) | ResNet-18 | SSOD (ResNet-18) |
> | --- | --- | --- | --- | --- | --- | --- | --- | --- |
> | Dataset | ImageNet-1K | ImageNet-1K | ImageNet-1K | ImageNet-1K | ImageNet-1K | ImageNet-1K | CIFAR-10 | CIFAR-10 |
> | ID ACC | 76.13 | 76.09 (-0.04%) | 77.37 | 76.92 (-0.45%) | 74.04 | 73.94 (-0.1%) | 94.21 | 94.17 (-0.04%) |
> - **Response to weakness 5**: In fact, concern about fairness is difficult to mitigate. In the comparison part, we collect some mainstream methods such as MSP, ODIN, GODIN, and Energy, and the latest methods such as MOS, Fort, KNN, and MCM. These methods are developed with some individual designs and architectures, and therefore, they are trained, finetuned, or post-processed in different ways. **We strictly keep the experimental settings identical for all comparable methods.** Besides, the fairness is quite difficult to evaluate. For example, ODIN resorts to pre-processing and post-processing without training the models, while SSOD resorts to finetuning without pre-processing or post-processing. We can not force all methods to follow the same manner to solve OOD detection. Nevertheless, we keep the experimental settings and evaluation protocol identical for all comparable methods.
> - **Response to minor points 1**: This bug appears if the PDF is opened using Adobe and disappears if using Google Chrome. We will fix this point.
> - **Response to minor points 2**: Yes, we agree with you and we are organizing the manuscript carefully. The content of `Appendix A.1` is vital since it interprets the existing OOD methods from our probabilistic view. We will move this part to the main body.

---

> > ### Comment · Reviewer_u8Y6 · 2023-11-15
> >
> > Thank you very much for your detailed response. I appreciate the time and effort you've put into addressing many of my concerns. I have a couple of points that I believe could benefit from further clarification to enhance my understanding:
> >
> > **Regarding Weakness 1:**
> >
> > a. I'd like to discuss the relationship between Eq. 7 and Eqs. 1-6. While it appears that Eq. 7 can be derived directly from Bayes's theorem, I'm curious about the specific necessity of Eqs. 1-6 in this context.
> >
> > b. I appreciate the efforts to provide a comprehensive perspective. However, I feel that Eq. 7, while insightful, essentially involves first determining if a sample is ID and then assessing the probability of it being an ID class. Eq. 7 does not necessarily deepen our understanding beyond the fundamental concept of OOD detection task. The proposed method actually does not reply on the unified view. The revisiting part in Appendix (A.1) uses the unified view, but the analyses can be done without the view. Could you shed more light on the pivotal role and benefits of adopting this unified view in your methodology?
> >
> > **Regarding Weakness 5:**
> >
> > Although the proposed method outperforms previous works, most of methods involved in the experiments are not directly comparable to the proposed method. Take Table 1 as an example.
> >
> > 1. MSP, MaDist, ODIN, Energy, KLM, KNN (w/o), and MCM are post-hoc methods assuming the access to a fixed pre-trained model. The proposed method needs to modify the training process of the model, so it is not surprising that it shows better performance than post-hoc methods.
> >
> > 2. MSP (CLIP-B/L), Fort (ViT-B/L), MCM (CLIP-B/L) are using different model architectures than others (using ResNet-50). This introduces more confounding factors.
> >
> > If you would like to refer to other benchmarks like OpenOOD [1,2], I suggest you could more focus on the comparison between the proposed method and other training methods on the same model architecture, such as SSD (also mentioned by another reviewer), CIDER, and LogitNorm. But currently, these methods are not included in the experiments.
> >
> >
> > [1] Yang, Jingkang, et al. "Openood: Benchmarking generalized out-of-distribution detection." Advances in Neural Information Processing Systems 35 (2022): 32598-32611.
> > [2] Zhang, Jingyang, et al. "OpenOOD v1. 5: Enhanced Benchmark for Out-of-Distribution Detection." arXiv preprint arXiv:2306.09301 (2023).

---

> > > ### Author Response · Authors · 2023-11-15
> > > **Response from the Authors**
> > >
> > > - **Response to weakness 1 (a)**: We emphasize that `Eq 7` holds with some conditions. Firstly, the $w_i$ represents some ID categories. Secondly, the ID data and OOD data are disjoint, enjoying no overlaps in data and label space. `Eqs 1-6` **explicitly** formulate the robust classification problem as conventional multi-category classification and OOD detection. `Eq 7` is the summarization and conclusion of `Eqs 1-6`. More concretely, without `Eqs 1-6`, we can not get the explicit formulation of robust classification. In `Eq 6`, the first term explicitly demonstrates the form of conventional softmax classification (i.e., the commonly used classification paradigm), while the second term tells us the gap between conventional classification and robust classification is OOD detection.
> > > - **Response to weakness 1 (b)**: `Eq 7` builds a formal description of robust classification, where OOD detection is one of the critical factors. Besides, we emphasize that current research in evaluating ID ACC also has some bias. In our experiments, we use $P(w_i|x)=P(w_i|x\in S_{ID}, x)P(x\in S_{ID}, x)$ to evaluate the ID ACC, which means if an ID image is identified as OOD, then it should be cast as the mis-classified sample. Moreover, `Eq 7` suggests that decoupling conventional classification and OOD detection may be a promising way to tackle robust classification, while most current methods mix these two tasks.
> > > - **Response to weakness 5**: We agree with you on this point. We will include more similar methods for comparison. We are familiar with OpenOOD and have gotten in touch with the authors to include SSOD as one of the technical proposals before the submission. We are preparing our codes to meet the format requirements of OpenOOD. We demonstrate part of the auxiliary comparison below. All values are FPR95 (lower is better).
> > > | Dataset (ID=ImageNet-1K) | iNaturalist | SUN | Places | Texture | Average |
> > > | --- | --- | --- | --- | --- | --- |
> > > | ReAct | 20.38 | 24.20 | 33.85 | 47.30 | 31.43 |
> > > | Dice | 25.63 | 35.15 | 46.49 | 31.72 | 34.75 |
> > > | Ash | 14.21 | 22.08 | 33.45 | 21.17 | 22.73 |
> > > | Ours (SSOD) | 14.80 | 28.52 | 38.92 | 45.32 | 31.89 |
> > >
> > >    | Dataset (ID=CIFAR-10) | SVHN | LSUN | iSUN | Texture | Places | Average |
> > >    | --- | --- | --- | --- | --- | --- | --- |
> > >    | ReAct | 41.64 | 11.46 | 12.72 | 43.58 | 43.31 | 30.54 |
> > >    | Dice | 25.99 | 3.91 | 4.36 | 41.90 | 48.59 | 24.95 |
> > >    | Ours (SSOD) | 2.12 | 4.42 | 10.06 | 1.91 | 7.44 | 5.19 |

---

> ### Comment · Reviewer_u8Y6 · 2023-11-21
>
> Thank you for your response. I appreciate the time and effort you have put into addressing my concerns.
>
> However, the significance of the probabilistic view is still unclear. According to your reply, I feel you are actually addressing Open-Set Recognition (OSR) [1], a task highly related to OOD detection. Notably, on page 3 of this work [1], there is a definition of open-set recognition that includes a two-stage prediction process for each sample: (i) assessing whether the test sample belongs to any known classes, and (ii) determining the distribution over the known classes. There are also metrics for OSR that may address the bias you mention in evaluating ID ACC. The existence of previous works does diminish the novelty of the proposed probabilistic view.
>
> Besides, regarding the benchmarks, it is beneficial to focus more on comparing the proposed method with other training methods within the same model architecture. The current experimental setup cannot prove the advantage of the proposed method, as I discussed in the previous comments.
>
> [1] "Open-Set Recognition: A Good Closed-Set Classifier is All You Need." ICLR 2022 Oral. https://openreview.net/forum?id=5hLP5JY9S2d

---

### Official Review · Reviewer_f34F · 2023-10-30

**Soundness:** 3 good
**Presentation:** 3 good
**Contribution:** 2 fair
**Rating:** 6
**Confidence:** 3

**Summary:**

The article focuses on Out-of-distribution (OOD) detection and proposes a model called Self-supervised Sampling for OOD Detection (SSOD) that does not require explicit OOD data annotation, which is able to extract natural OOD signals from the background of ID images, end-to-end end-to-end training of OOD detection branches. Experiments are conducted on several large-scale OOD detection datasets to demonstrate the effectiveness and superiority of SSOD.

**Strengths:**

1.	The article proposes a general and reasonable probabilistic framework to understand the OOD detection problem, which can cover a wide range of existing OOD detection methods in an innovative way.
2.	The article proposes an effective self-supervised sampling mechanism capable of extracting useful OOD signals from ID images, avoiding the difficulty and cost of collecting and labeling large amounts of OOD data.
3.	The article provides a detailed description and derivation of the working principle and design of SSOD.

**Weaknesses:**

1.	The article uses some uncritical and unreasonable assumptions in the derivation of the probabilistic framework, such as setting T as sM+1, ignoring the possible differences between sM+1 and T; treating P(wi|x) as P(wi|x∈SID,x), ignoring the possibility that x may belong to the OOD data.
2.	The article uses a fixed and subjective threshold γ in the self-supervised sampling mechanism to determine whether an image block belongs to ID or OOD, which does not consider the possible differences and variations between different datasets, models, and categories.
3.	There are some spelling mistakes, grammatical errors, and punctuation errors in the article; some irregular or inappropriate terms are used in the article, such as OOD-data-free model, OOD patch sampler, and so on.

**Questions:**

1.	The derivation of formula (1) seems to lack a detailed explanation. Can a more complete mathematical derivation of this formula be provided?
2.	There are some grammatical and spelling errors in the text, please fix them
3.	There seems to be a subjective bias in the interpretation of the experimental results. Can more objective evidence be provided to support these interpretations?
4.	In the first paragraph on page 1, the author mentions that "OOD detection empowers the model trained on the closed image set to identify unknown data in the open world". But there is no definition or difference between what is closed image set and open world. Please define and explain these two concepts clearly in the introduction.
5.	On page 4, paragraph 5, the author mentioned "Since the ImageNet-O mainly contains adversarial images, leading to the classifier's wrong prediction, SSOD reports higher FPR95 compared to the open world". SSOD reports higher FPR95 compared to the best previous methods", but no reason or mechanism is given as to why the adversarial images cause SSOD's performance to degrade.

---

> ### Author Response · Authors · 2023-11-13
> **Response from the Authors**
>
> Thanks for the valuable reviews from Reviewer **#f34F**. We detail the response as follows.
> - **Response to weakness 1**: I have to argue that using $T$ as $s_{M+1}$ is not an assumption, it is just an equivalent substitution. As depicted in `Eq 1`, both $s_i$ and $T$ are trainable parameters of neural networks, and what we do in `Eq 5` is just use $s_{M+1}$ to indicate $T$. Moreover, we argue that treating $P(w_i|x)$ as $P(w_i|x\in S_{ID}, x)$ is not an assumption as well. Based on the bayes' theorem, we have:
>
>      $P(w_i|x)=P(w_i|x\in S_{ID}, x) P( x \in S_{ID}|x) + P(w_i|x\in S_{OOD}, x) P(x\in S_{OOD}|x)$.
>      In Section 3.1, we have highlighted that $w_i$ indicates the ID categories, and $S_{ID}$ has no overlap with $S_{OOD}$. Therefore, the term $P(w_i|x\in S_{OOD}, x)$ equals zero.
> - **Response to weakness 2**: SSOD performs consistently well across whole benchmarks using $\gamma$ as 0.95. We manually try several different values of $\gamma$ on different benchmarks and find this parameter is stable. Due to the limited space, we didn't provide this part in ablation. Please refer to our response to **Reviewer #ajFn**, where we provide detailed ablations on $\gamma$. The more detailed experimental results on these parameters will be attached in our revised Appendix.
> - **Response to weakness 3**: We will correct these errors in our revised manuscript.
> - **Response to question 1**: In Eq 1, $\sigma(\cdot)$ indicates the `Sigmoid` function, and $g_i(x)=-s_i+T$ is a score function. More concretely, $g_i(x)=-s_i(x)+T$, where both $s_i(x)$ and $T$ can be regarded as trainable neural networks. $s_i(x)$ changes with the input $x$. $T$ is the learnable bias term which is a constant. Therefore, we can use $T$ as $s_{M+1}$ since they are both unchanged with the input image $x$. The details about `Sigmoid` function can be found here: https://en.wikipedia.org/wiki/Sigmoid_function.
> - **Response to question 2**: We will correct these errors in our revised manuscript.
> - **Response to question 3**: We compare all methods based on valid experiments, besides, for those results from the original copy, we highlight them as well. For example, in `Section 4.2`, we detail the evaluation protocol and the experimental settings. All conclusions are drawn from the experimental results. **We admire all of these investigated methods which have promoted the development of OOD detection and robust classification**. Our team has no subjective judgment during the evaluation procedure. We will refine the description and look forward to your detailed reply.
> - **Response to question 4**: The closed image set indicates the image dataset with a fixed number of categories. For example, the ImageNet-1K is a closed image dataset with 1000 categories. For those objects not included within this scope, a conventional classification model trained on ImageNet-1K fails to identify them. The open world indicates the real world where any kind of image can appear. These terminologies usually appear in the field of **Open World Recognition**, and we will present a more detailed description. Besides, this survey [1] shows the overall view of OOD detection and robust classification, which may help to mitigate this concern.
> - **Response to question 5**: The failure of SSOD in adversarial images is led by incorrect supervision from the CLS head. Recall that the mechanism of SSOD is joint learning between the CLS head and the OOD head. Adversarial images lead to poor performance of the CLS head, and therefore, the OOD head encounters a performance drop as well. More concretely, in `Eq 9`, the poor CLS head will provide incorrect confidence $\hat{y}_i^{HW}$ for the OOD head.
>
> [1] Jingkang Yang, Kaiyang Zhou, Yixuan Li, and Ziwei Liu. Generalized Out-of-Distribution Detection: A Survey, 2022

---

### Official Review · Reviewer_TJ6P · 2023-10-30

**Soundness:** 3 good
**Presentation:** 3 good
**Contribution:** 3 good
**Rating:** 6
**Confidence:** 5

**Summary:**

- Authors tackle the problem of Out-Of-Distribution (OOD) detection in this work and show that the image background in In-Distribution (ID) datasets can act as a good proxy for OOD data, preventing the necessity to collect real/synthetic OOD data for training strong OOD detectors.
- First, authors propose a general probabilistic framework that can explain existing OOD detection methods.
- Next, using this interpretation, authors propose **S**elf-**S**upervised **O**OD **D**etection (SSOD) to exploit the natural OOD signals present in ID data. This prevents the need for collecting explicit real or synthetic OOD data making the pipeline more efficient.
- With extensive experiments, authors show impressive results on several OOD benchmarks proving that image backgrounds of ID data can serve as a good proxy for OOD data.

**Strengths:**

- The probabilistic interpretation of OOD detection methods is very useful to advance future research. The factorized interpretation also helps tune each independent component accordingly to optimize ID or OOD performance depending on the downstream objective.
- The paper is well written, and the math is easy to follow once derived on paper.
- The experimental section supports all the claims made in the paper.

**Weaknesses:**

I will summarize my concerns with this work under three broad sections.

**Nomenclature**
- Authors chose to proceed with the name SSOD for their work but the whole field of Semi-Supervised Object detection (SSOD) [1] already exists creating a bit of a confusion.
- I recommend using SSOOD to avoid any confusion with an already established sub-field.

**Presentation of results**
- Authors claim that their first contribution is to provide a probabilistic interpretation of OOD, using which existing methods can be analyzed, but pushed the analysis section to the supplementary. In my opinion, if authors claim the probabilistic interpretation as an analysis tool, then it shouldn't be delegated to the supplementary.

**Motivation and intuition**
- Authors show impressive results on several OOD benchmarks but the motivation that image background can serve as a good proxy for OOD data has some flaws.
- First, using the penguin image example from Fig. 2, teaches the network to consider the background (in this case "water") as a signal for OOD. Now this will be ineffective in datasets constructed from comics or cartoons which is not one of the domains that authors evaluate their method on. This also explains why the scores are lower on OOD datasets constructed from SUN, Places because Imagenet is predominantly biased towards "organisms" and "food" with lower signals from indoor scenes and places. This raises the question "Does this method work because of the choice of the OOD datasets used?" This is partly explained by the results on "Textures" (I agree with the authors that textures has some overlap with patterns in the ID classes and the results are low, but I believe that is just part of the story).
- Second, I think some additional analysis is required on the iNaturalist dataset (or maybe another toy setup on the subset of imagenet) which explains why the method works. If the OOD head is learning to detect any background patch as OOD, then what is the role of this in iNaturalist, where the background is usually water/trees/nature etc? From Imagenet, the network learnt to flag any background containing these regions as OOD, so does it ignore the ID category entirely and just focus on background? But that can't be true because the ID performance is also higher. The interplay between the CLS head and OOD head is very important to completely understand why image background is a good proxy and is missing from the paper.

**I recommend authors to consider answering these questions for me to improve on my rating**

[1] Liu, Yen-Cheng and Ma, Chih-Yao and He, Zijian and Kuo, Chia-Wen and Chen, Kan and Zhang, Peizhao and Wu, Bichen and Kira, Zsolt and Vajda, Peter, Unbiased Teacher for Semi-Supervised Object Detection, ICLR 2021.

**Questions:**

**Questions**
- In the abstract, authors mention they leverage "local property of convolution" for OOD. But it hasn't been mentioned anywhere else. Can you elaborate what they mean by this?
- In eq. 9 the $y_i^{\text{OOD}}$ is the label for detecting OOD patches right? In which case, patches with confidence lower than 5% ($1-\gamma$) should have a label 1 and not 0 right?
- In the 2nd sentence below Eq. 11, is it "During inference" or "During training/inference"? The second loss term in Eq. 11 is applied on a spatial map from what I understood, so why do we have to compute $P(x\in \mathcal{S}_{\mathbb{ID}}|x)$ explicitly during training? We just need that during inference correct?

**Details Of Ethics Concerns:**

I do not foresee any immediate ethical concerns with this work.

---

> ### Author Response · Authors · 2023-11-12
> **Response from the Authors**
>
> The suggestions from Reviewer **#TJ6P** are quite valuable and well-judged. We provide our response in detail as follows.
> - **Response to nomenclature**: Thanks a lot for your suggestion and the better name. Honestly, we omit this point during the writing of our manuscript. We chose SSOD as the name of our method only because of its suitable meaning, i.e., self-supervised sampling for OOD detection. We want to emphasize the self-supervised manner and the OOD proxy sampling. Your suggested SSOOD is clearer and more suitable since it indicates the sub-field of OOD detection. We will modify the method name in our revised manuscript.
> - **Response to the presentation of results**: The probabilistic framework of OOD detection is one of our main contributions. We detail this part in `Section 3.1`, however, the interpretation of existing OOD methods from this view is attached in `Appendix A.1`. This is quite unsuitable. We are organizing our manuscript currently, and we plan to move a part of the related work and experimental setup to the Appendix and transfer the interpretation part from the Appendix to the main body. We have to admit that the organization of this manuscript is a little hurried. Thanks again for your suggestions for improving our manuscript.
> - **Response to motivation and intuition 1**: SSOOD explores the image background as the OOD proxies, providing OOD supervision during the training of the binary classification head. Sailing from this fundamental premise, SSOOD encounters failure most in the following two cases. Firstly, the target object of the input image carries similar characters as the background of ImageNet. Secondly, the learned backgrounds in ImageNet can not cover the background of input images, for example, different domains or styles. Obviously, you fully have the core of SSOOD, including its superiority and limitations. An effective way to close this gap is to train the SSOOD with sufficient and diverse images. In our in-house application, we pre-train the SSOOD on ImageNet-1K and continue to train it on several combined datasets collected from our practical scenarios. The data from the target scenario significantly guarantees the performance of outlier detection.
> - **Response to motivation and intuition 2**: This is quite an amazing problem. I am trying to figure out your concern from my limited understanding. Firstly, the training of robust classification consists of OOD detection (binary classification) and conventional multi-category classification as shown in `Eq 11`, and therefore, the classification accuracy gets marginal influence. Besides, within the loss of the OOD head, we collect both ID patches (`objects in ImageNet, labeled as 1`) and OOD patches (`backgrounds in ImageNet, labeled as 0`) to supervise the training, thus the model will not ignore the ID category. In ImageNet vs. iNaturalist, the backgrounds of ImageNet usually appear like some lakes, trees, or nature, and the OOD head treats this information as OOD characters. During the inference, the trained OOD head will assign lower confidence (`note: lower confidence from the OOD head indicates a higher probability that the input is OOD data`) to iNaturalist since it is similar to the background of ImageNet. I doubt that I only get a part of your problem, and I am looking forward to your reply and continuing this discussion. Again, quite admire your insights.
> - **Response to question 1**: The local property of convolution indicates the limited perceptive field of conventional neural networks. For example, using the ResNet-50 to extract the feature of an input image in the shape of 224x224x3, the yielded feature is in the shape of 2048x7x7, i.e., 7x7 patch features whose dimension is 2048. The local property indicates that the top-left (the first) feature patch only consists of local information from the top-left part of the image instead of global information. And that is the critical factor that SSOOD works. Still, we are extending SSOOD to self-attention-based models, and the key problem is how to extract pure background information since the self-attention mechanism fuses all image patches such as ViT and SwinTransformer.
> - **Response to question 2**: We design the OOD head as a binary classification head. Higher confidence outputted by the OOD head indicates a higher probability the input is ID data, i.e., **the ID data is labeled as 1 while the OOD data is labeled as 0 by the OOD head.** We term this head as OOD head may incur some misunderstanding. **Shortly, the output confidence of the OOD head reflects the probability that the input belongs to ID.** For example, if the output confidence is 0, then the input is OOD data; and if the output confidence is 1, then the input is quite likely ID data.
> - **Response to question 3**: Right. This term (OOD factor) is only needed during the inference phase.

---

> > ### Comment · Reviewer_TJ6P · 2023-11-15
> > **Response to Author's comments**
> >
> > I thank the author for their detailed response to all my concerns and overall I'm happy with the response and am willing to increase my score. Please see a few more comments below.
> > **Clarification to Response to motivation and intuition 2**: The background information in iNaturalist and most animal classes in imagenet is very similar (trees, lakes, nature etc.). So if iNaturalist dataset is used as the OOD dataset, even if there are some classes overlapping between the ID dataset and OOD dataset, the model will learn to flag any image with nature in the background as OOD. Maybe this can be better seen by creating a subset within imagenet. Ideally we want the model to look at the foreground object within an image to decide if its ID or OOD. But in this case, SSOOD is heavily relying on the background. Hope that makes sense.
> >
> > **Response to question 1**: For ViTs maybe a masked attention can potentially be helpful as it avoids a patch to attend to everything else.
> >
> > **Response to question 2**: If its the OOD head we are talking about, then doesn't it make sense to assign label 1 to OOD and 0 to ID?

---

> > > ### Author Response · Authors · 2023-11-16
> > > **Response from the Authors**
> > >
> > > We respond to the proposed questions in order.
> > > - **Response 1**: Thanks for your clarification, and we are very clear that we have fully got your point. **Your concern is that the model may learn a shortcut to identify all images as OOD if they have similar backgrounds in ImageNet.** For example, SSOOD performs better on iNaturalist, and this phenomenon is potentially caused by the similarity between the backgrounds of ImageNet and iNaturalist. Besides, you suggest that SSOOD should identify whether an image is ID or OOD mainly based on the foreground objects instead of focusing more on the backgrounds. This topic is quite vital and needs far more investigation and verification. Currently, we will perform an experiment that employs a subset (for example, we only use the categories of vegetables, clothing, and vehicles) of ImageNet as ID data to train the model from scratch. With this trained model, we evaluate it on iNaturalist where the image backgrounds are different. We naturally expect that the FPR95 of SSOOD is lower than MSP under this setting. However, it may be higher than the model trained on the whole ImageNet-1K. Your suggestion inspires us that there still exists a lot of work to fill, for example, how to guide the model to focus on the foreground objects solely instead of learning shortcuts from the backgrounds. On the other hand, we also emphasize the necessity of image backgrounds, since they play the role of OOD supervision. **An important topic is to balance the dependence degree on foreground objects and backgrounds during the ID/OOD identification.**
> > > - **Response 2**: Yes, we also considered this manner. At present, we forward the input image (224x224x3) to the linear projector and obtain 196 tokens. We randomly sample spatially continuous tokens and add an OOD token ahead of them. The following procedure is similar to that of CNN models. Honestly, we only reach a part of promising results on this part, marginally better than ResNet, while we also encounter some failure cases. The length of tokens (default is 196, if we perform sampling, the length will decrease) also influences the result. We are still investigating this topic and hope we will find something helpful to other researchers.
> > > - **Response 3**: Thanks for your suggestion. We will refine the name of the OOD head. This appears to cause some misunderstanding for most researchers.
> > >
> > > I appreciate the insights and inspirations I learned during the discussion.

---

> > > > ### Comment · Reviewer_TJ6P · 2023-11-22
> > > > **Response to Author's comments**
> > > >
> > > > I thank the reviewer for their effort. I agree that more work is needed to understand why solely background helps detecting OOD samples and not the foreground object in the OOD images. Since this is extremely important to appreciate the contributions, I would like to keep my original rating.

---

### Official Review · Reviewer_ajFn · 2023-11-01

**Soundness:** 3 good
**Presentation:** 3 good
**Contribution:** 3 good
**Rating:** 6
**Confidence:** 2

**Summary:**

This paper focuses on Out-of-Distribution (OOD) detection and introduces a unified probabilistic framework that divides the OOD detection problem into In-Distribution (ID) and OOD components. This framework presents an insightful overview of existing OOD detection methodologies and pinpoints their limitations (classifiers and features are often biased towards ID data). To address this challenge, the authors introduce Self-Supervised Sampling for OOD Detection (SSOD), utilizing image backgrounds as effective proxies for OOD data. The model employs separate ID and OOD heads, with the OOD head being self-trained through the utilization of confidence scores derived from the classification head. The results demonstrate that the proposed method significantly outperforms existing approaches by a substantial margin.

**Strengths:**

1. The motivation derived from the proposed general OOD detection framework seems both intuitive and solid.
2. The paper is well-written and figures / tables are easy to follow.
3. The proposed SSOD approach, while seemingly simple, demonstrates strong effectiveness.
4. The experimental results are quite strong across various datasets.

**Weaknesses:**

1. Given that SSOD necessitates pseudo-labels for each patch (like semantic segmentation),
Since SSOD requires the pseudo-labels for each patch (like semantic segmentation), I presume that the training expenses could surpass those of conventional pre-training methods. Could the authors provide a computational comparison of SSOD with other baseline OOD detection models, as well as standard classification models (e.g., ResNet-18, ResNet-50, etc.)?

2. It appears that the principal interpretation derived from the probabilistic framework (Appendix A.1) might be more aptly positioned between Sections 3.1 and 3.2. Currently, the main motivation behind SSOD doesn't seem to be adequately emphasized.

**Questions:**

1. Could you specify the number of patches used in the SSOD model? More specifically, what are the dimensions (height H and width W) in the last feature map?

2. Does the performance of SSOD show sensitivity to the classification confidence parameter gamma?

3. While SSOD generally surpasses other baselines in performance, as seen in the tables, there are instances where other baselines demonstrate notably strong results (e.g., MOS in iNaturalist and KNN in Texture). What could be the main reason for these exceptional cases?

---

> ### Author Response · Authors · 2023-11-12
> **Response from the Authors**
>
> Thanks for the valuable suggestions from Reviewer **#ajFn**. We demonstrate the response below.
> - **Response to weakness 1**: We provide the parameters and MACs (Multiplication and Accumulation) of different models as follows. The evaluation code is from: https://github.com/Lyken17/pytorch-OpCounter, and all input images are in the shape of 224x224x3, and the number of classes is set to 1000. We will update the results of other methods ASAP.
> |  Model Name  | ResNet-18 | SSOD (ResNet-18) | ResNet-50 | SSOD (ResNet-50) |
> |  ----  | ----  |  ----  | ----  |  ----  |
> |  Params (M) | 11.69 | 12.40 | 25.56 | 28.19 |
> | MACs (G) | 1.82 | 1.93 | 4.14 | 4.57 |
> - **Response to weakness 2**: We will move a part of the content from Appendix A.1 to the main body of the manuscript, better illustrating our motivation.
> - **Response to question 1**: The dimensions of the feature map depend on the employed backbone and the input images. For example, using a 224x224x3 image and ResNet-50 backbone, the yielded feature map is in the shape of 2048x7x7, i.e., the feature channel is 2048 and the number of patches is 49. The number of employed patches is adaptive as demonstrated in Figure 2 and Eq 9. For instance, in Figure 2 (d), the patches in red/green are used as OOD/ID. The number of employed patches is about 45% of the overall image.
> - **Response to question 2**: SSOD is stable. In our experiments, we set the $\gamma$ from 0.8 to 0.99 with a step of 0.05. Using the ImageNet vs. iNaturalist as the benchmark, and the ResNet-50 as the backbone. The detailed results are demonstrated below. Since the threshold may be different with the changing backbones, we don't detail this part for its marginal significance.
> | $\gamma$ | 0.8 | 0.85 | 0.9 | 0.95 | 0.99 |
> | --- | --- | --- | --- | --- | --- |
> | FPR95 | 18.64 | 17.52 | 15.88 | 14.80 | 15.62 |
> - **Response to question 3**: The performance of different methods depends on both the distribution of the dataset and the parameter space of the model itself. To provide a convincing result, we test comparable methods on several benchmarks and compute the averaged performance. For example, in Table 1, the overall FPR95 and AUROC of SSOD are 1.89% and 91.26%, surpassing the following method with -6.28% FPR95 and +0.77% AUROC. SSOD reports competitive performance (however, poorer than KNN) on Texture, that is because Texture overlaps with ImageNet, we have demonstrated this part in Appendix A.5 (Failure Case Analyses). We suggest that KNN is a clustering-based method. Maybe in the Texture dataset, its yielded manifold and classification boundary are more compact than our discriminative SSOD, therefore, achieving the best FPR95 compared to current approaches.

---

> > ### Comment · Reviewer_ajFn · 2023-11-22
> >
> > Thanks for the response from the authors.
> >
> > Considering the overall quality of the paper, I hold my original score.

---

### Official Review · Reviewer_f65W · 2023-11-05

**Soundness:** 4 excellent
**Presentation:** 4 excellent
**Contribution:** 3 good
**Rating:** 8
**Confidence:** 3

**Summary:**

The paper addresses the task of out-of-distribution detection in image classification tasks. The authors identify two main challenges the existing methods are facing, namely the absence of a unified perspective to interpret existing techniques and the need for natural OOD supervision to enhance model robustness without the necessity of explicit OOD data collection. Towards this end, the authors introduce a novel probabilistic framework that provides a unified interpretation of many existing OOD detection methods and the SSOD model that efficiently leverages the CNNs properties (retain spatial information) to create a distinction between ID and OOD samples. The paper's contributions are substantiated through extensive experiments (both is terms of benchmarks, as well as proper evaluation w.r.t state-of-the-art methods). SSOD shows consistent improvements in OOD detection metrics over previous methods, highlighting the model's effectiveness in recognizing and handling OOD samples without the need for additional OOD data.

**Strengths:**

Originality >> SSOD uses image backgrounds as natural proxies for OOD samples, which is a novel perspective in the field.

Quality >> The best paper I read in a while, both in terms of problem statement/formulation and execution (exemplar experimental analysis).

Clarity >> The paper is well-articulated - clearly presents the research problem, the proposed solution, and the insights from the conducted experiments. Also, the authors seem to have made an effort to ensure that the concepts are accessible to anyone, regardless of their expertise in OOD detection.

Significance >> By proposing a general probabilistic framework, the paper unifies various existing approaches under a single interpretative lens, on top of SSOD that offers an evident advancement for OOD detection (relevant problem).

**Weaknesses:**

No fundamental flaws with the current submission, but just a suggestion to the authors - to include a more thorough discussion on the limitations of SSOD - what are the potential biases, the impact of background complexity on the model's performance, and scenarios where the model may not perform as expected - to start a discussion towards further improvements. Maybe within a dedicated section, tackling these aspects would be valuable.

**Questions:**

No further questions, besides addressing the weaknesses mentioned above.

---

> ### Author Response · Authors · 2023-11-12
> **Response from the Authors**
>
> Firstly, we are honored to receive such high praise from Reviewer **#f65W**, and thanks for the valuable suggestions. In our current manuscript, the limitation analyses on SSOD is attached in  `Appendix A.5` (Failure Case Analyses). And in our revised manuscript, we will extend the part **Conclusion and Discussion** to include more insights on broadening the scope of SSOD. We have noticed the potential of extending SSOD to transformer architectures, and the partly finished experiments surprised us. There are many details of generalizing SSOD to self-attention-based models. For instance, how to sample local background given the fact that self-attention fuses global information, and how to perform inference since the tokens of background patches are less than the whole images. All these works will be finished ASAP and demonstrated on Arxiv. **Besides, we here want to highlight that SSOD has been deployed in our in-house application.** The engineers highly praise the transferability and scalability of SSOD since it is quite easy to deploy as an end-to-end model.

---

### Author Response · Authors · 2023-11-18
**Response to all reviewers**

**Dear all reviewers**:

We thank all reviewers for your very constructive suggestions and well-judged reviews to improve our manuscript. Up to now, we have responded to all concerns proposed in the former stage. **We sincerely suggest the reviewers follow these responses and continue further discussions. Feel free to ask more questions or change your rating.** We are looking forward to your reply.

---

### Meta-Review · Area_Chair_vj7z · 2023-12-03

**Metareview:**

The paper proposes a probabilistic framework to achieve OOD detection from the ID data without collecting explicit OOD samples. The proposed method frames the OOD detection problem as an image classification problem and leverages the signals collected from the background variations in the ID data.

Overall, four out of six reviewers provided a positive review and think that the proposed probabilistic framework for OOD detection from the ID data is novel, and the method achieves state-of-the-art (SOTA) results. The comments from these reviewers were addressed by the authors during the rebuttal process, and the reviewers were satisfied with the authors' responses.

Two reviewers, Reviewer EtXA and Reviewer u8Y6, were not in favor of the paper. Reviewer u8Y6 thinks that the proposed method is novel and achieves state-of-the-art results. However, Reviewer u8Y6 had some major concerns regarding the probabilistic formulation of the problem, the confidence threshold, and experimental bias. The reviewer acknowledged the authors' response to their initial comments. The reviewer also commented on the similarity of the problem to the open-set recognition problem but didn't receive a response from the authors. Reviewer u8Y6 thinks that the proposed method has the potential to impact further research on the topic and it achieves state-of-the-art results. The reviewer had some concerns regarding the experimental results and experiments with additional methods. The authors addressed the reviewer's concerns, which I found satisfactory; however, they didn't receive a response from the reviewer.

Given the above discussion and rebuttal/changes to the paper, I recommend acceptance. It is a very well-done paper, provides interesting findings, stronger baselines, and thorough experimentation. Furthermore, the probabilistic formulation of learning OOD signals from ID data is a valuable contribution to the community.

**Justification For Why Not Higher Score:**

The rebuttal didn't considerably alter the reviewers' rating.

**Justification For Why Not Lower Score:**

The probabilistic formulation of learning OOD signals from ID data is a valuable contribution to the community.

---

### Decision · Program_Chairs · 2024-01-16

Accept (poster)